# Federated Learning with Manifold Regularization and Normalized Update Reaggregation

**Xuming An**[1]    **Li Shen**[2*]    **Han Hu**[1*]    **Yong Luo**[3]
[1] School of Information and Electronics, Beijing Institute of Technology, China
[2] JD Explore Academy, China    [3] School of Computer Science, Wuhan University, China
{anxuming,hhu}@bit.edu.cn, mathshenli@gmail.com, luoyong@whu.edu.cn

## Abstract

Federated Learning (FL) is an emerging collaborative machine learning framework where multiple clients train the global model without sharing their own datasets. In FL, the model inconsistency caused by the local data heterogeneity across clients results in the near-orthogonality of client updates, which leads to the global update norm reduction and slows down the convergence. Most previous works focus on eliminating the difference of parameters (or gradients) between the local and global models, which may fail to reflect the model inconsistency due to the complex structure of the machine learning model and the Euclidean space's limitation in meaningful geometric representations. In this paper, we propose FedMRUR by adopting the manifold model fusion scheme and a new global optimizer to alleviate the negative impacts. Concretely, FedMRUR adopts a hyperbolic graph manifold regularizer enforcing the representations of the data in the local and global models are close to each other in a low-dimensional subspace. Because the machine learning model has the graph structure, the distance in hyperbolic space can reflect the model bias better than the Euclidean distance. In this way, FedMRUR exploits the manifold structures of the representations to significantly reduce the model inconsistency. FedMRUR also aggregates the client updates norms as the global update norm, which can appropriately enlarge each client's contribution to the global update, thereby mitigating the norm reduction introduced by the near-orthogonality of client updates. Furthermore, we theoretically prove that our algorithm can achieve a linear speedup property $\mathcal{O}(\frac{1}{\sqrt{SKT}})$ for non-convex setting under partial client participation, where $S$ is the participated clients number, $K$ is the local interval and $T$ is the total number of communication rounds. Experiments demonstrate that FedMRUR can achieve a new state-of-the-art (SOTA) accuracy with less communication.

## 1   Introduction

FL is a collaborative distributed framework where multiple clients jointly train the model with their private datasets [27, 28]. To protect privacy, each client is unable to access the other dataset [2]. A centralized server receives the parameters or gradients from the clients and updates the global model[46]. Due to the limited communication resource, only part of the clients is involved in the collaborative learning process and train the local model in multiple intervals with their own datasets within one communication round [23]. Due to the data heterogeneity, clients' partial participation and multiple local training yield severe model inconsistency, which leads to the divergences between the directions of the local updates from the clients and thus reduces the magnitude of global updates [17]. Therefore, the model inconsistency is the major source of performance degradation in FL [40, 15].

---

[*]Corresponding authors: Li Shen and Han Hu

37th Conference on Neural Information Processing Systems (NeurIPS 2023).

So far, numerous works have focused on the issues of model inconsistency to improve the performance of FL. Many of them [20, 16, 42, 1] utilize the parameter (or gradient) difference between the local and global model to assist the local training. By incorporating the global model information into local training, the bias between the local and global objectives can be diminished at some level. However, the parameter (or gradient) difference may fail to characterize the model bias due to the complex structure of modern machine learning model and the Euclidean space has limitations in providing powerful and meaningful geometric representations [10]. Meanwhile, incorporating the difference introduces extra high computation and communication costs because of the high-dimensional model parameter, which is common in the modern machine learning area[25]. Some other works [22, 39, 44, 24] exploit the permutation invariance property of the neurons in the neural networks to align and aggregate the model parameters for handling the model inconsistency issues, but the extra computation required for neuron alignment may slow down the speed of FL. In addition, Charles *et al.* [3] demonstrate that after multiple rounds, the similarities between the client updates approach zero and the local update direction are almost orthogonal to each other in FL. If the server aggregates the local updates, each client's contribution is little, which reduces the global update step. Therefore, we need to reduce the model inconsistency and compensate for the global norm reduction introduced by the near-orthogonality of client updates.

In order to alleviate the model inconsistency and compensate for the global reduction, we propose a practical and novel algorithm, dubbed as **FedMRUR** (**Fed**erated learning with **M**anifold **R**egularization and Normalized **U**pdate **R**eaggregation). FedMRUR adopts two techniques to achieve SOTA performance. i) Firstly, FedMRUR adopts the hyperbolic graph fusion technique to reduce the model inconsistency between the client and server within local training. The intuition is that adding the manifold regularizer to the loss function to constrain the divergence between the local and global models. Unlike the Euclidean space, the hyperbolic space is a manifold structure with the constant negative curvature, which has the ability to produce minimal distortion embedding[8] with the low storage constraints[30] for graph data. And the neural network, the most prevail machine learning model, has a graph structure[34, 24], we map the representations to the hyperbolic space and compute their distance to indicate the model bias precisely. Considering the numerical stability[6], we select the Lorentz model to describe the hyperbolic space and the squared Lorentzian distance[19] to indicate the representations' proximity. By adopting the hyperbolic graph fusion technique, FedMRUR can constrain model inconsistency efficiently. ii) Secondly, FedMRUR aggregates the client's local updates in a novel normalized way to alleviate the global norm reduction. In the normalized aggregation scheme, the server aggregates the local update norms as the global norm and normalizes the sum of the local updates as the global direction. Compared with directly aggregating local updates, the new aggregation scheme enables each customer's contribution to be raised from its projection on the global direction to its own size. As a result, the size of the global update becomes larger and compensates for the norm reduction introduced by model inconsistency, which improves the convergence and generalization performance of the FL framework.

Theoretically, we prove that the proposed FedMRUR can achieve the convergence rate of $\mathcal{O}(\frac{1}{\sqrt{SKT}})$ on the non-convex and L-smooth objective functions with heterogeneous datasets. Extensive experiments on CIFAR-10/100 and TinyImagenet show that our proposed FedMRUR algorithm achieves faster convergence speed and higher test accuracy in training deep neural networks for FL than several baselines including FedAvg, FedProx, SCAFFOLD, FedCM, FedExp, and MoFedSAM. We also study the impact on the performance of adopting the manifold regularization scheme and normalized aggregation scheme. In summary, the main contributions are as follows:

- We propose a novel and practical FL algorithm, FedMRUR, which adopts the hyperbolic graph fusion technique to effectively reduce the model inconsistency introduced by data heterogeneity, and a normalized aggregation scheme to compensate the global norm reduction due to the *near-orthogonality* of client updates, which achieves fast convergence and generalizes better.

- We provide the upper bound of the convergence rate under the smooth and non-convex cases and prove that FedMRUR has a linear speedup property $\mathcal{O}(\frac{1}{\sqrt{SKT}})$.

- We conduct extensive numerical studies on the CIFAR-10/100 and TinyImagenet dataset to verify the performance of FedMRUR, which outperforms several classical baselines on different data heterogeneity.

## 2 Related Work

McMahan *et al.* [27] propose the FL framework and the well-known algorithm, FedAvg, which has been proved to achieve a linear speedup property [43]. Within the FL framework, clients train local models and the server aggregates them to update the global model. Due to the heterogeneity among the local dataset, there are two issues deteriorating the performance: the model biases across the local solutions at the clients [20] and the similarity between the client updates (which is also known as the *near-orthogonality* of client updates) [3], which needs a new aggregation scheme to solve. In this work, we focus on alleviating these two challenges to improve the convergence of the FL algorithms.

**Model consistency.** So far, numerous methods focus on dealing with the issue of model inconsistency in the FL framework. Li *et al.* [20] propose the FedProx algorithm utilizing the parameter difference between the local and global model as a prox-correction term to constrain the model bias during local training. Similar to [20], during local training, the dynamic regularizer in FedDyn [1] also utilizes the parameter difference to force the local solutions approaching the global solution. FedSMOO [36] utilizes a dynamic regularizer to make sure that the local optima approach the global objective. Karimireddy *et al.* [16] and Haddadpour *et al.* [9] mitigate the model inconsistency by tracking the gradient difference between the local and global side. Xu *et al.* [42] and Qu *et al.* [31] utilize a client-level momentum term incorporating global gradients to enhance the local training process. Sun *et al.* [37] estimates the global aggregation offset in the previous round and corrects the local drift through a momentum-like term to mitigate local over-fitting. Liu *et al.* [26] incorporate the weighted global gradient estimations as the inertial correction terms guiding the local training to enhance the model consistency. Charles *et al.* [4] demonstrate that the local learning rate decay scheme can achieve a balance between the model inconsistency and the convergence rate. Tan *et al.* [38] show that the local learning rate decay scheme is unable to reduce the model inconsistency when clients communicate with the server in an asynchronous way. Most methods alleviate the model inconsistency across the clients by making use of the parameter (or gradient) difference between the local and global model.

**Aggregation scheme.** There are numerous aggregation schemes applied on the server side for improving performance. Some works utilize classical optimization methods, such as SGD with momentum [45], and adaptive SGD [5], to design the new global optimizer for FL. For instance, FedAvgM [13, 35] and STEM [17] update the global model by combining the aggregated local updates and a momentum term. Reddi *et al.* [32] propose a federated optimization framework, where the server performs the adaptive SGD algorithms to update the global model. FedNova [41] normalizes the local updates and then aggregates them to eliminate the data and device heterogeneity. In addition, the permutation invariance property of the neurons in the neural networks is also applied for improving robustness to data heterogeneity. FedFTG [48] applies the data-free knowledge distillation method to fine-tune the global model in the server. FedMA [39] adopts the Bayesian optimization method to align and average the neurons in a layer-wise manner for a better global solution. Li *et al.* [22] propose Position-Aware Neurons (PANs) coupling neurons with their positions to align the neurons more precisely. Liu *et al.* [24] adopt the graph matching technique to perform model aggregation, which requires a large number of extra computing resources in the server. Many deep model fusion methods [21] are also applied in the research field of FL, such as model ensemble [47] and CVAE [12]. The aforementioned algorithms utilize the local parameters or gradients directly without considering the *near-orthogonality* of client updates, which may deteriorate the convergence performance of the FL framework.

The proposed method FedMRUR adopts the hyperbolic graph fusion technique to reduce the model inconsistency and a normalized update aggregation scheme to mitigate the norm reduction of the global update. Compared with the previous works, we utilize the squared Lorentzian distance of the features in the local and global model as the regularization term. This term can more precisely measure the model bias in the low-dimensional subspace. For the update aggregation at the server, FedMRUR averages the local updates norm as the global update norm, which achieves to alleviate the norm reduction introduced by the near-orthogonality of the client updates.

## 3 Methodology

In this section, we first formally describe the problem step for FL and then introduce the FedMRUR and the two novel hyperbolic graph fusion and normalized aggregation techniques in FedMRUR.

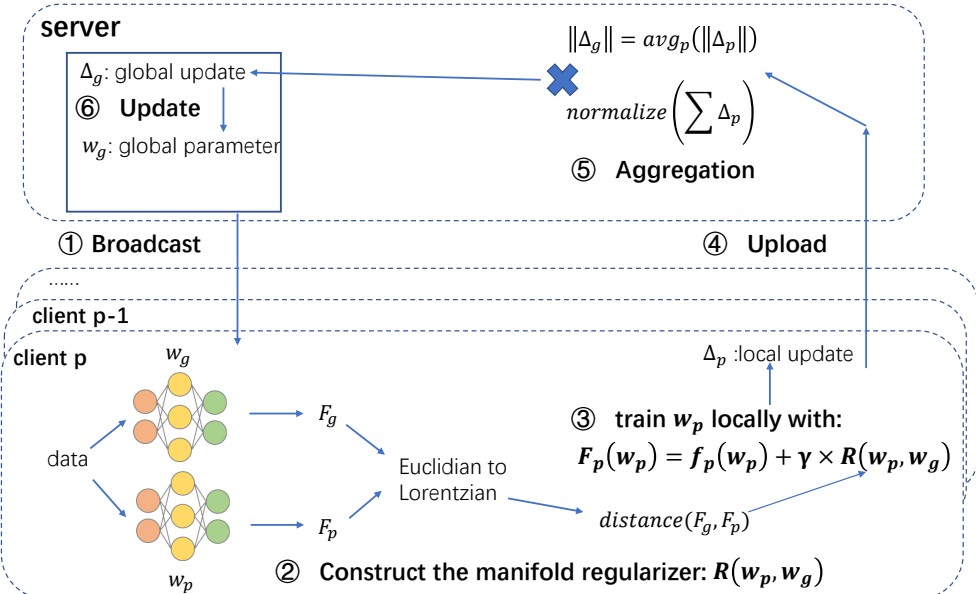

Figure 1: The workflow of FedMRUR. Once the global parameter $x_0$ is received, the client initializes the local model $x_p$ and starts hyperbolic graph fusion. In the hyperbolic graph fusion, the client first takes the local and global model to get their representations and maps them into the hyperbolic space. Then, the client use their distances in the hyperbolic space as a regularizer to constrain model divergence. Next, the client performs local training and uploads the updates to the server. The server adopts the normalized scheme to aggregate the local updates and performs the global model update.

## 3.1   Problem setup

We consider collaboratively solving the stochastic non-convex optimization problem with $P$ clients :

$$\min_w f(w) := \frac{1}{P} \sum_{p \in \mathcal{P}} f_p(w), \text{ with } f_p(w) := \mathbb{E}_{z \sim D_p}[l(w, z)], \tag{1}$$

where $w$ is the machine learning model parameter and $z$ is a data sample following the specified distribution $D_p$ in client $p$; meanwhile $l(w, z)$ represents the model loss function evaluated by data $z$ with parameter $w$. $f_p(w)$ and $f(w)$ indicate the local and global loss function, respectively. The loss function $l(w, z)$, $f_p(w)$, and $f(w)$ are non-convex due to the complex machine learning model. The heterogeneity of distribution $D_p$ causes the model inconsistency across the clients, which may degrade the performance of the FL framework.

**Notations.** We define some notations to describe the proposed method conveniently. $\| \cdot \|$ denotes the spectral norm for a real symmetric matrix or $L_2$ norm for a vector. $\langle \cdot, \cdot \rangle$ denotes the inner product of two vectors. For any nature $a$, $b$, $a \wedge b$ and $a \vee b$ denote $\min \{a, b\}$ and $\max \{a, b\}$, respectively. The notation $O(\cdot)$, $\Theta(\cdot)$, and $\Omega(\cdot)$ are utilized to hide only absolute constants that don't depend on any problem parameter.

## 3.2   FedMRUR Algorithm

In this part, we describe our proposed FedMRUR algorithm (see Figure 1 and Algorithm 1) to mitigate the negative impacts of model inconsistency and improve performance. We add a manifold regularization term on the objective function to alleviate the model inconsistency. To eliminate the near-orthogonality of client updates, we design a new method to aggregate the local updates from the clients. Within one communication round, the server first broadcast the global model to the participating clients. During local training, the client takes the sampled data into the local and received global model and gets the representations. Then the client maps the representations into the hyperbolic space and computes their distance, which is used to measure the divergence between the local and global models. Next, the client adopts the distance as a manifold regular to constrain the model bias, achieving model fusion in the hyperbolic graph. After local training, the client uploads its local update to the server. The server aggregates the local update norms as the global update step

**Algorithm 1** FedMRUR

1: **Input:** initial parameter $w^0$; momentum coefficient $\alpha$; perturbation radius $\rho$; local interval $K$ ; communication rounds $T$; set of selected clients $S_t$; global and local learning rate $\eta_g, \eta_l$.
2: **Output:** Global parameter $w^t, \forall t \in T$.
3: **Initialization:** Initialize $\triangle^0 = \mathbf{0}$ and $w^0$ as the global parameter at the server.
4: For $t = 0, 1, ..., T-1$ do:
5:      The server broadcasts parameter $w^t$ and global update $\triangle^t$ to the selected clients $S_t$.
6:      For client $p \in S_t$ in parallel do:
7:      client $p$ initialize the local parameter as $w_p^{t,0} = w^t$.
8:          For $k = 0, ..., K-1$ do:
9:              $\widetilde{w}_p^{t,k} = w_p^{t,k} + \rho \frac{\nabla F_p(w_p^{t,k})}{\|\nabla F_p(w_p^{t,k})\|}$.
10:             $v_i^{t,k+1} = \alpha \nabla F_p(\widetilde{w}_p^{t,k}) + (1-\alpha)\triangle^t$.
11:             $w_i^{t,k+1} = w_i^{t,k} - \eta_l v_i^{t,k+1}$.
12:          End for.
13:          $\triangle_p^t = w_p^{t,K} - w_p^{t,0}$
14:      End for
15:      Aggregate $\triangle^{t+1} = \frac{\sum_{p \in S_t} \|\triangle_p^t\|}{|S_t| \| \sum_{p \in S_t} \triangle_p^t \|} \sum_{i \in S_t} \triangle_p^t$.
16:      Update global parameter $w^{t+1} = w^t - \eta_g \triangle^{t+1}$.
17: End for.

and normalizes the sum of the local updates as the global update direction. Utilizing the normalized aggregation scheme, the server can update the model with a larger step and improve the convergence.

**Hyperbolic Graph Fusion.** In FL, the Euclidean distances between parameters[11, 20] (or gradients[16, 42]) between the client and the server is utilized to correct the local training for alleviating the model inconsistency. However, the Euclidean distance between the model parameters can't correctly reflect the variation in functionality due to the complex structure of the modern machine learning model. The model inconsistency across the clients is still large, which impairs the performance of the FL framework. Since the most prevail machine learning model, neural network has a graph structure and the hyperbolic space exhibits minimal distortion in describing data with graph structure, the client maps the representations of the local and global model into the hyperbolic shallow space[29] and uses the squared Lorentzian distance[19] between the representations to measure the model inconsistency.

To eliminate the model inconsistency effectively, we adopt the hyperbolic graph fusion technique, adding the distance of representations in the hyperbolic space as a regularization term to the loss function. Then the original problem (1) can be reformulated as:

$$\min_{w_0} F(w_0) = \frac{1}{P} \sum_p [f_p(w_p) + \gamma * R(w_p, w_g)], \quad s.t. \ w_g = \frac{1}{P} \sum_p w_p \qquad (2)$$

where $R(w_p, w_g)$ is the hyperbolic graph fusion regularization term, defined as:

$$R(w_p, w_g) = \exp\left(\|L_p - L_g\|_{\mathcal{L}}^2/\sigma\right), \quad \|L_p - L_g\|_{\mathcal{L}}^2 = -2\beta - 2\langle L_p, L_g\rangle_{\mathcal{L}}. \qquad (3)$$

In (2) and (3), $L_p$ and $L_g$ are the mapped Lorentzian vectors corresponding to $Z_p$ and $Z_g$, the representations from local model $w_p$ and global model $w_g$. $\gamma$ and $\sigma$ are parameters to tune the impact of the model divergence on training process. $\beta$ is the parameter of the Lorentz model and $\langle x, y\rangle_{\mathcal{L}}$ denotes the Lorentzian scalar product defined as:

$$\langle x, y\rangle_{\mathcal{L}} = -x_0 \cdot y_0 + \sum_{i=1}^d x_i \cdot y_i, \qquad (4)$$

where $x$ and $y$ are $d+1$ dimensional mapped Lorentzian vectors. The new problem (2) can be divided into each client and client $p$ uses its local optimizer to solve the following sub-problem:

$$\min_{w_p} F_p(w_p) = f_p(w_p) + \gamma * R(w_p, w_g). \qquad (5)$$

The regularization term $R(w_p, w_g)$ has two benefits for local training: (1) It mitigates the local over-fitting by constraining the local representation to be closer to the global representation in the hyperbolic space (Lorentzian model); (2) It adopts representation distances in a low-dimensional hyperbolic space to measure model deviation, which can be more precisely and save computation.

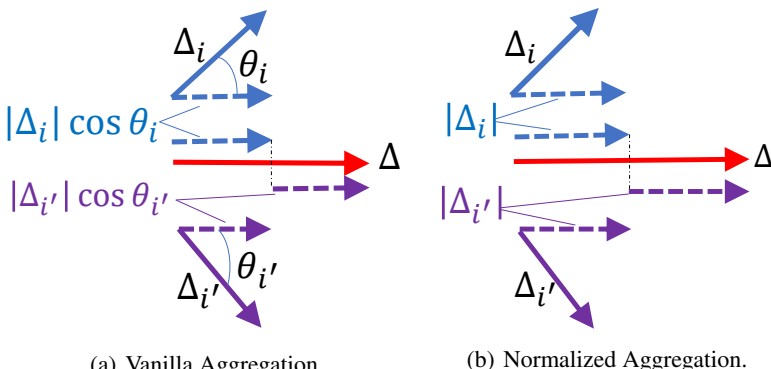

(a) Vanilla Aggregation.      (b) Normalized Aggregation.

Figure 2: A toy schematic to compare the naive aggregation and normalized aggregation of the local updates, where the number of clients is 2 and the local intervals are set as 1. The solid line indicates the client's local update $\triangle_i$, $\theta_i$ is the angle between the local update and the global update, and the dotted line represents the clients' contribution on the global update. The red lines are the aggregated global update $\triangle$. The main difference is $\|\triangle\|$, the norm of the global update. When adopting the naive aggregation method, the global norm $\|\triangle\| = \sum_i \|\triangle_i\| \cos \theta_i$. When adopting the normalized aggregation method, the global norm $\|\triangle\| = \sum_i \|\triangle_i\|$. We can see that the norm of global update in the direct aggregation is less than the norm in the normalized aggregation, due to the fact that $\cos \theta \leq 1$.

**Normalized Aggregation of Local Updates.** According to [3], after a number of communication rounds, the cosine similarities of the local updates across clients are almost zero, which comes from the model(gradient) inconsistency between the server and client sides. In the vanilla aggregation way, the server takes the mean of local updates from participated clients as the global gradient. As shown in Figure 2(a), client $i$ makes $\|\triangle_i\| \cos \theta_i$ contribution on the aggregation result. When the data heterogeneous is significant, the cosine similarities across the local updates are small. Correspondingly, the cosine similarities $\cos \theta_i$ between the clients and the server narrows down, and the global update norm shrinks which slows down the convergence. To alleviate the negative impact of near-orthogonal local updates, we propose a new normalized aggregation method to compute the global update. The direction of the global update can be acquired by normalizing the sum of the local updates and the result is the same as the one obtained by the vanilla way. For the norm, the server computes it by taking the average of norms of the received local updates. As shown in Figure 2(b), with the proposed normalized aggregation method, the client $i$'s contribution on the global update increases from $\|\triangle_i\| \cos \theta_i$ to $\|\triangle_i\|$. Accordingly, the norm of the global update $\|\triangle\|$ grows and accelerates the convergence.

Our proposed FedMRUR is characterized by Figure 1 and the detailed training process is summarized in Algorithm 1. Concretely, firstly, the server broadcasts the global parameter and updates it to the selected clients $S_t$. At the initialization stage of local training, client $i$ utilizes the manifold regularization scheme to construct its own local loss function $F_i$ with the received global parameter $w^t$. Then, client $i$ adopts the Sharpness Aware Minimization (SAM) [7] optimizer to compute the gradient $\widetilde{g}_i$ with data sampled randomly. The local updater $v_i$ consists of the stochastic gradient $\alpha \widetilde{g}_i$ and the momentum term $(1 - \alpha)\triangle^t$, the received global update from the last round. Client $i$ applies $v_i$ to perform multiple SGD and uploads the accumulated local update $\triangle_i^t$ to the server. The server takes two steps to construct the global update: 1) aggregating and normalizing the accumulated local updates from the participated clients $S_t$ as the direction of the global update; 2) averaging the norms of accumulated local updates as the norm of the global update. Finally, the server utilizes the constructed global update to perform one step SGD and get a new global parameter.

**Remark 1.** *FedMRUR is on the top of MoFedSAM [31] due to its excellent performance and our method can also be integrated with other federated learning methods, including FedExp, FedCM, SCAFFOLD, FedDYN, etc., to improve the performance.*

# 4 Convergence Analysis

In this section, we provide the theoretical analysis of our proposed FedMRUR for general non-convex FL setting. Due to space limitations, the detailed proofs are placed in **Appendix**. Before introducing the convergence results, we first state some commonly used assumptions as follows.

**Assumption 1.** *$f_p(x)$ is L-smooth and $R(x, x_0)$ is r-smooth with fixed $x_0$ for all client p, i.e.,*

$$\|\nabla f_p(a) - \nabla f_p(b)\| \leq L \|a - b\|, \quad \|\nabla R(a, x_0) - \nabla R(b, x_0)\| \leq r \|a - b\|.$$

**Assumption 2.** *The stochastic gradient $g_p^{t,k}$ with the randomly sampled data on the local client p is an unbiased estimator of $\nabla F_p(x_p^{t,k})$ with bounded variance, i.e.,*

$$E[g_p^{t,k}] = \nabla F_p(x_p^{t,k}), \ E \left\| g_p^{t,k} - \nabla F_p(x_p^{t,k}) \right\|^2 \leq \sigma_l^2$$

**Assumption 3.** *The dissimilarity of the dataset among the local clients is bounded by the local and global gradients, i.e.,*

$$E \|\nabla F_p(x) - \nabla F(x)\|^2 \leq \sigma_g^2 \tag{6}$$

Assumption 1 guarantees the gradient Lipschitz continuity for the objective function and regularizer term. Assumption 2 guarantees the stochastic gradient is bounded by zero mean and constant variance. Assumption 3 gives the heterogeneity bound for the non-iid dataset across clients. All the above assumptions are widely used in many classical studies [1, 43, 33, 42, 14, 16], and our convergence analysis depends on them to study the properties of the proposed method.

**Proof sketch.** To explore the essential insights of the proposed FedMRUR, we first bound the client drift over all clients within the t-th communication round. Next, we characterize the global parameter moving within a communication round, which is similar to the one in centralized machine learning algorithms with momentum acceleration technology. Then, the upper bound for the global update $\triangle_t$ is provided. Lastly, we use $\|\nabla F(x_t)\|$ the global gradient norm as the metric of the convergence analysis of FedMRUR. The next theorem characterizes the convergence rate for FedMRUR.

**Theorem 1.** *Let all the assumptions hold and with partial client participation. If $\eta_l \leq \frac{1}{\sqrt{30}\alpha K L}$, $\eta_g \leq \frac{S}{2\alpha L(S-1)}$ satisfying $\frac{3}{4} - \frac{2(1-\alpha L)}{KN} - 70(1-\alpha)K^2(L+r)^2\eta_l^2 - \frac{90\alpha(L+r)^3\eta_g\eta_l^2}{S} - \frac{3\alpha(L+r)\eta_g}{2S}$, then for all $K \geq 0$ and $T \geq 1$, we have:*

$$\frac{1}{\sum_{t=1}^{T} d_t} \sum_{t=1}^{T} \mathbb{E} \left\| \nabla F(w^t) \right\|^2 d_t \leq \frac{F^0 - F^*}{C\alpha\eta_g \sum_{t=1}^{T} d_t} + \Phi, \tag{7}$$

*where*

$$\Phi = \frac{1}{C} \Big[ 10\alpha^2(L+r)^4\eta_l^2\rho^2\sigma_l^2 + 35\alpha^2 K(L+r)^2\eta_l^2 3(\sigma_g^2 + 6(L+r)^2\rho^2) + 28\alpha^2 K^3(L+r)^6\eta_l^4\rho^2$$

$$+ 2K^2 L^4\eta_l^2\rho^2 + \frac{\alpha(L+r)^3\eta_g^2\rho^2}{2KS}\sigma_l^2 + \frac{\alpha(L+r)\eta_g}{K^2 SN}(30NK^2(L+r)^4\eta_l^2\rho^2\sigma_l^2$$

$$+ 270NK^3(L+r)^2\eta_l^2\sigma_g^2 + 540NK^2(L+r)^4\eta_l^2\rho^2 + 72K^4(L+r)^6\eta_l^4\rho^2$$

$$+ 6NK^4(L+r)^2\eta_l^2\rho^2 + 4NK^2\sigma_g^2 + 3NK^2(L+r)^2\rho^2) \Big].$$

*and $d_t = \frac{\sum_i \|\triangle_i^t\|}{\|\sum_i \triangle_i^t\|} \geq 1$. Specifically, we set $\eta_g = \Theta(\frac{\sqrt{SK}}{\sqrt{T}})$ and $\eta_l = \Theta(\frac{1}{\sqrt{ST}K(L+r)})$, the convergence rate of the FedMRUR under partial client participation can be bounded as:*

$$\sum_{t=1}^{T} E \|\nabla F(x_t)\|^2 = O\left(\frac{1}{\sqrt{SKT}}\right) + O\left(\frac{\sqrt{K}}{ST}\right) + O\left(\frac{1}{\sqrt{KT}}\right). \tag{8}$$

**Remark 2.** *Compared with the inequality $\frac{F^0 - F^*}{C\alpha\eta_g T} + \Phi$ of Theorem D.7 in MoFedSAM paper[31], the second constant term in (7) is same and the first term is less than the first term in MoFedSAM paper, which validates FedMRUR achieves faster convergence than MoFedSAM.*

**Remark 3.** *From (8), we can find that when $T$ is large enough, the dominant term $O(\frac{1}{\sqrt{SKT}})$ in the bound achieves a linear speedup property with respect to the number of clients. It means that to achieve $\epsilon-$precision, there are $O(\frac{1}{SK\epsilon^2})$ communication rounds required at least for non-convex and L-smooth objective functions.*

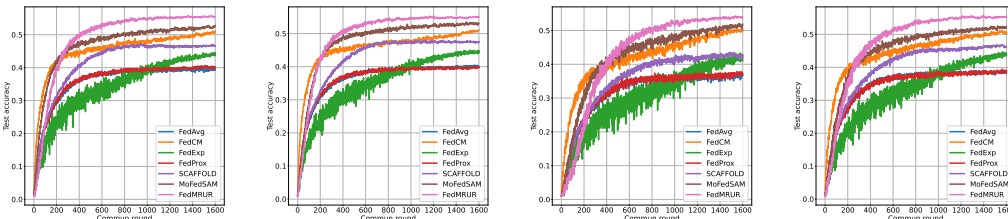

(a) Test accuracy on CIFAR-100 in the non-iid ($\mu = 0.3$, $\mu = 0.6$, $n = 10$ and $n = 20$) settings.

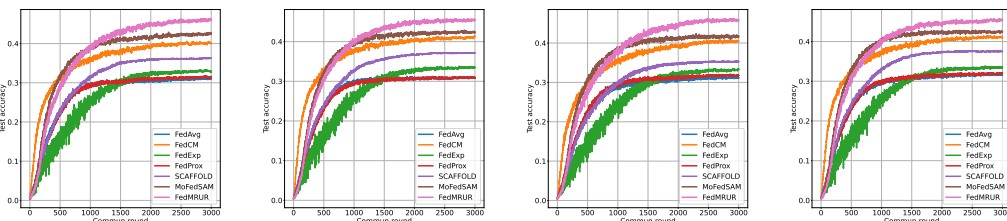

(b) Test accuracy on TinyImageNet in the non-iid ($\mu = 0.3$, $\mu = 0.6$, $n = 40$ and $n = 80$) settings.

Figure 3: Test accuracy w.r.t. communication rounds of our proposed method and other approaches. Each method performs in 1600 communication rounds. To compare them fairly, the basic optimizers are trained with the same hyperparameters.

## 5 Experiments

In this section, we validate the effectiveness of the proposed FedMRUR algorithm using the experimental results on CIFAR-10/100 [18] and TinyImageNet [18]. We demonstrate that FedMRUR outperforms the vanilla FL baselines under heterogeneous settings. We also present that both manifold regularization and the proposed normalized update aggregation can improve the performance of SGD in FL. The experiments of CIFAR-10 are placed in the **Appendix.**

### 5.1 Experimental Setup

**Datasets.** We compare the performance of FL algorithms on CIFAR-10/100 and TinyImageNet datasets with 100 clients. The CIFAR-10 dataset consists of $50K$ training images and $10K$ testing images. All the images are with $32 \times 32$ resolution belonging to 10 categories. In the CIFAR-100 dataset, there are 100 categories of images with the same format as CIFAR-10. TinyImageNet includes 200 categories of $100K$ training images and $10K$ testing images, whose resolutions are $64 \times 64$. For non-iid dataset partitioning over clients, we use Pathological-$n$ (abbreviated as Path($n$)) and Dirichlet-$\mu$ (abbreviated as Dir($\mu$)) sampling as [13], where the coefficient $n$ is the number of data categories on each client and $\mu$ measures the heterogeneity. In the experiments, we select the Dirichlet coefficient $\mu$ from $\{0.3, 0.6\}$ for all datasets and set the number of categories coefficient $n$ from $\{3, 6\}$ on CIFAR-10, $\{10, 20\}$ on CIFAR-100 and $\{40, 80\}$ on TinyImageNet.

**Implementation Details.** For all algorithms on all datasets, following [1, 42], the local and global learning rates are set as $0.1$ and $1.0$, the learning rate decay is set as $0.998$ per communication round and the weight decay is set as $5 \times 10^{-4}$. ResNet-18 together with group normalization is adopted as the backbone to train the model. The clients' settings for different tasks are summarized in Table 1. Other optimizer hyperparameters are as follow: $\rho = 0.5$ for SAM, $\alpha = 0.1$ for client momentum, $\gamma = 0.005$, $\sigma = 10000.0$ and $\beta = 1$ for manifold regularization.

Table 1: The experiments settings for different tasks.

| Task | num of clients | participated ratio | batch size | local epoch |
|------|---------------|--------------------|-----------|-------------|
| CIFAR | 200 | 0.05 | 50 | 3 |
| Tiny | 500 | 0.02 | 20 | 2 |

**Baselines.** To compare the performances fairly, the random seeds are fixed. We compare the proposed FedMRUR with several competitive benchmarks: FedAvg [43], the most widely used baseline,

Table 2: Test accuracy (%) on CIFAR-100& TinyImagenet datasets in both Dir($\mu$) and Path($n$)) distributions.

| Algorithm | CIFAR-100 | | | | TinyImagenet | | | |
|---|---|---|---|---|---|---|---|---|
| | Dir($\mu$)) | | Path($n$) | | Dir($\mu$)) | | Path($n$) | |
| | $\mu=0.6$ | $\mu=0.3$ | n=20 | n=10 | $\mu=0.6$ | $\mu=0.3$ | n=80 | n=40 |
| FedAvg | 39.87 | 39.50 | 38.47 | 36.67 | 30.78 | 30.64 | 31.62 | 31.18 |
| FedExp | 44.51 | 44.26 | 43.58 | 41.00 | 33.49 | 32.68 | 33.65 | 33.39 |
| FedProx | 39.89 | 39.86 | 38.82 | 37.15 | 30.93 | 31.05 | 32.09 | 31.77 |
| SCAFFOLD | 47.51 | 46.47 | 46.23 | 42.45 | 37.14 | 36.22 | 37.48 | 35.32 |
| FedCM | 51.01 | 50.93 | 50.58 | 50.03 | 41.37 | 40.21 | 40.93 | 40.46 |
| MoFedSAM | 52.96 | 52.81 | 52.32 | 51.87 | 42.36 | 42.29 | 42.52 | 41.58 |
| FedMRUR | 55.81 | 55.49 | 55.21 | 53.69 | 45.54 | 45.41 | 45.42 | 45.71 |

Table 3: Convergence speed comparison on CIFAR100& TinyImageNet datasets. "Acc." represents the target test accuracy on the dataset. "$\infty$" means that the algorithm is unable to achieve the target accuracy on the dataset.

| Datasets | CIFAR-100 | | | | | TinyImageNet | | | | |
|---|---|---|---|---|---|---|---|---|---|---|
| Algorithms | Acc. | Dir($\mu$) | | Path($n$) | | Acc. | Dir($\mu$) | | Path($n$) | |
| | | 0.6 | 0.3 | 20 | 10 | | 0.6 | 0.3 | 80 | 40 |
| FedAvg | | 513 | 494 | 655 | $\infty$ | | 972 | 1078 | 1002 | 1176 |
| FedExp | | 715 | 782 | 795 | 1076 | | 1255 | 1362 | 1327 | 1439 |
| FedProx | | 480 | 488 | 638 | $\infty$ | | 1043 | 1030 | 1163 | 1615 |
| SCAFFOLD | 38% | 301 | 322 | 389 | 585 | 30% | 785 | 850 | 766 | 967 |
| FedCM | | 120 | 126 | 157 | 255 | | 342 | 401 | 366 | 474 |
| MoFedSAM | | 154 | 146 | 211 | 300 | | 436 | 447 | 415 | 460 |
| Our | | 157 | 179 | 223 | 341 | | 473 | 517 | 470 | 570 |
| FedAvg | | $\infty$ | $\infty$ | $\infty$ | $\infty$ | | $\infty$ | $\infty$ | $\infty$ | $\infty$ |
| FedExp | | 985 | 1144 | 1132 | 1382 | | $\infty$ | $\infty$ | $\infty$ | $\infty$ |
| FedProx | | $\infty$ | $\infty$ | $\infty$ | $\infty$ | | $\infty$ | $\infty$ | $\infty$ | $\infty$ |
| SCAFFOLD | 42% | 406 | 449 | 558 | 998 | 35% | 1289 | 1444 | 1206 | 2064 |
| FedCM | | 173 | 193 | 260 | 527 | | 599 | 735 | 674 | 879 |
| MoFedSAM | | 197 | 192 | 260 | 392 | | 598 | 624 | 583 | 685 |
| Our | | 192 | 230 | 266 | 424 | | 671 | 707 | 653 | 788 |
| FedAvg | | $\infty$ | $\infty$ | $\infty$ | $\infty$ | | $\infty$ | $\infty$ | $\infty$ | $\infty$ |
| FedExp | | $\infty$ | $\infty$ | $\infty$ | $\infty$ | | $\infty$ | $\infty$ | $\infty$ | $\infty$ |
| FedProx | | $\infty$ | $\infty$ | $\infty$ | $\infty$ | | $\infty$ | $\infty$ | $\infty$ | $\infty$ |
| SCAFFOLD | 45% | 521 | 616 | 784 | $\infty$ | 40% | $\infty$ | $\infty$ | $\infty$ | $\infty$ |
| FedCM | | 276 | 353 | 470 | 842 | | 1451 | 2173 | 1587 | 2186 |
| MoFedSAM | | 243 | 278 | 400 | 575 | | 950 | 1106 | 959 | 1162 |
| Our | | 241 | 263 | 338 | 484 | | 948 | 1050 | 953 | 1069 |

firstly applies local multiple training and partial participation for FL framework; SCAFFOLD [16] utilizes the SVRG method to mitigate the client drift issue; FedProx [20] uses a proximal operator to tackle data heterogeneity; FedCM [42] incorporates the client-momentum term in local training to maintain the model consistency among clients; Based on FedCM, MoFedSAM [31] improves the generalization performance with local SAM [7] optimizer; FedExp [14] determines the server step size adaptively based on the local updates to achieve faster convergence.

## 5.2 Evaluation Results

Figure 3 and Table 2 demonstrate the performance of ResNet-18 trained using multiple algorithms on CIFAR-100 and TinyImageNet datasets under four heterogeneous settings. We plot the test accuracy of the algorithms for a simple image classification task in the figure. We can observe that: our proposed FedMRUR performs well with good stability and effectively alleviates the negative impact of the model inconsistency. Specifically, on the CIFAR100 dataset, FedMRUR achieves 55.49% on the Dirichlet-0.3 setups, which is 5.07% higher than the second-best test performance. FedMRUR effectively reduces the model inconsistency and the enlarged global update improves the speed of convergence.

Table 3 depicts the convergence speed of multiple algorithms. From [13], a larger $\mu$ indicates less data heterogeneity across clients. We can observe that: 1) our proposed FedMRUR achieves the fastest convergence speed at most of the time, especially when the data heterogeneity is large. This

validates that FedMRUR can speed up iteration; 2) when the statistical heterogeneity is large, the proposed FedMRUR accelerates the convergence more effectively.

## 5.3 Ablation Study

**Impact of partial participation.** Figures 4(a) and 4(b) depict the optimization performance of the proposed FedMRUR with different client participation rates on CIFAR-100, where the dataset splitting method is Dirichlet sampling with coefficient $\mu = 0.3$ and the client participation ratios are chosen from 0.02 to 0.2. From this figure, we can observe that the client participation rate (PR) has a positive impact on the convergence speed, but the impact on test accuracy is little. Therefore, our method can work well under low PR settings especially when the communication resource is limited.

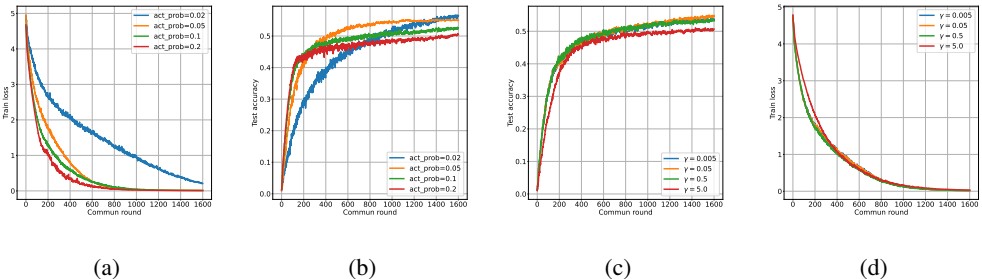

(a)        (b)        (c)        (d)

Figure 4: (a). Training loss w.r.t different client participation rations; (b). Test accuracy w.r.t different client participation ratios. (c). Test accuracy with different $\gamma$. (d). Train loss with different $\gamma$. The performance of FedMRUR with different parameters on the CIFAR-100 dataset.

**Hyperparameters Sensitivity.** In Figures 4(c) and 4(d), we compare the performance of the proposed FedMRUR with different hyper-parameters on the CIFAR-100 dataset. From the results, we can see that our algorithm achieves similar test accuracy and training loss under different $\gamma$ within a certain range ($\gamma \in [0.005, 0.5]$) and this indicates the proposed FedMRUR is insensitive to the hyperparameter $\gamma$. The hyperparameter $\gamma$ represents the method the punishment on the model bias.

**Impact of Each Component.** Table 4 demonstrates the impact of each component of FedMRUR on the test accuracy for CIFAR-100 dataset on the Dirichlet-0.3 setups. For convenience, we abbreviate normalized aggregation as "normalized(N)" and hyperbolic graph fusion as "hyperbolic(H)", respectively. From the results, we can find that both the normalized local update

Table 4: Test accuracy % on CIFAR-100 datasets about without each ingredients of FedMRUR.

| Algorithm | normalized(N) | hyperbolic(H) | Acc. |
|---|---|---|---|
| MoFedSAM | – | – | 52.81 |
| FedMRUR-N | ✓ | – | 54.27 |
| FedMRUR-H | – | ✓ | 53.57 |
| FedMRUR | ✓ | ✓ | 55.49 |

aggregation scheme and the hyperbolic graph fusion can improve performance. This table validates that our algorithm design and theoretical analysis are correct and effective.

## 6 Conclusion

In this work, we propose a novel and practical federated method, dubbed FedMRUR which applies the hyperbolic graph fusion technique to alleviate the model inconsistency in the local training stage and utilizes normalized updates aggregation scheme to compensate for the global norm reduction due to the *near-orthogonality* of the local updates. We provide the theoretical analysis to guarantee its convergence and prove that FedMRUR achieves a linear-speedup property of $O(\frac{1}{\sqrt{SKT}})$. We also conduct extensive experiments to validate the significant improvement and efficiency of our proposed FedMRUR, which is consistent with the properties of our analysis. This work inspires the FL framework design to focus on exploiting the manifold structure of the learning models.

**Limitations&Broader Impacts.** Our work focuses on the theory of federated optimization and proposes a novel FL algorithm. During the local training, the representations of the global model must be stored locally, which may bring extra pressure on the client. This will help us in inspiration for new algorithms. Since FL has wide applications in machine learning, Internet of Things, and UAV networks, our work may be useful in these areas.

**Acknowledgements.** This work is supported by National Key Research and Development Program of China under SQ2021YFC3300128, and National Natural Science Foundation of China under Grant 61971457. Thanks for the support from CENI-HEFEI and Laboratory for Future Networks in University of Science and Technology of China.

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
