convergence analysis for the proposed FedMRUR algorithm. Firstly, we state some preliminary lemmas as follows:

**Lemma 2.** *For random variables $x_1, ..., x_n$, we have*

$$\mathbb{E}\left[\|x_1 + ... + x_n\|^2\right] \leq n\mathbb{E}[\|x_1\|^2 + ... + \|x_n\|^2].$$

**Lemma 3.** *For independent, mean $0$ random variables $x_1, ..., x_n$, we have*

$$\mathbb{E}\left[\|x_1 + ... + x_n\|^2\right] = \mathbb{E}[\|x_1\|^2 + ... + \|x_n\|^2]$$

**Lemma 4.** *The stochastic gradient $\nabla F_i(w, \xi_i)$ computed by the i-th client at model parameter $w$ using minibatch $\xi_i$ is an unbiased estimator of $\nabla F_i(w)$ with variance bounded by $\sigma^2$. The gradient of SAM is formulated by*

$$\mathbb{E}\left[\left\|\sum_{k=0}^{K-1} g_i^{t,k}\right\|^2\right] \leq K \sum_{k=0}^{K-1} \mathbb{E}\left[\left\|\nabla F_i(w_i^{t,k})\right\|^2\right] + \frac{K(L+r)^2\rho^2}{N}\sigma_l^2$$

*Proof.* we can bound the inequality as follows:

$$\mathbb{E}\left[\left\|\sum_{k=0}^{K-1} g_i^{t,k}\right\|^2\right] = \mathbb{E}\left[\left\|\sum_{k=0}^{K-1} \nabla F_i(w_i^{t,k})\right\|^2\right] + \mathbb{E}\left[\left\|\sum_{k=0}^{K-1} g_i^{t,k} - \nabla F_i(w_i^{t,k})\right\|^2\right]$$

$$\leq K \sum_{k=0}^{K-1} \mathbb{E}\left[\left\|\nabla F_i(w_i^{t,k})\right\|^2\right] \tag{9a}$$

$$+ (L+r)^2 \sum_{k=0}^{K-1} \mathbb{E}\left[\frac{1}{N}\sum_{i=1}^{N}\left(w_i^{t,k} + \delta_i^{t,k}(\tilde{w}_i^{t,k}; \xi_i^{t,k}) - w_i^{t,k} + \delta_i^{t,k}(\tilde{w}_i^{t,k})\right)\right]$$

$$\leq K \sum_{k=0}^{K-1} \mathbb{E}\left[\left\|\nabla F_i(w_i^{t,k})\right\|^2\right] + \frac{K\rho^2\sigma_l^2}{N}(L+r)^2 \tag{9b}$$

where (9a) is from Assumption 1 and (9b) is from Assumption 3 and Lemma 3. $\square$

**Lemma 5.** *The variance of local and global gradients with perturbation can be bounded as follows:*

$$\|\nabla F_i(w + \delta_i) - \nabla F(w + \delta)\|^2 \leq 3\sigma_g^2 + 6(L+r)^2\rho^2.$$

*Proof.*

$$\|\nabla F_i(\tilde{w}) - \nabla F(\tilde{w})\|^2 = \|\nabla F_i(w + \delta_i) - \nabla F(w + \delta)\|^2$$

$$= \|\nabla F_i(w + \delta_i) - \nabla F_i(w) + \nabla F_i(w) - \nabla F(w) + \nabla F(w) - \nabla F(w + \delta)\|^2$$

$$\leq 3\|\nabla F_i(w + \delta_i) - \nabla F_i(w)\|^2 + 3\|\nabla F_i(w) - \nabla F(w)\|^2 + 3\|\nabla F(w) - \nabla F(w + \delta)\|^2 \tag{10a}$$

$$\leq 3\sigma_g^2 + 6(L+r)^2\rho^2, \tag{10b}$$

where (10a) is from Lemma 2 and (10b) is from Assumption 1,2, the perturbation is limited by $\rho$. $\square$

Below, we bound the average client drift over all clients within the $t-$ communication round. The average client drift is bounded by

**Lemma 6.** *Given $\eta_l \leq \frac{1}{\sqrt{30}\alpha K(L+r)}$ and $\alpha \leq \frac{1}{2}$, there is*

$$
\begin{aligned}
\epsilon_{t,k} =& \frac{1}{|S_t|} \sum_{i \in S_t} \mathbb{E} \left\| w_i^{t,k} - w^t \right\|^2 \\
\leq& 5K\eta_l^2 \left[ 2\alpha^2(L+r)^2\eta_l^2\rho^2\sigma_l^2 + 7K\alpha^2\eta_l^2(3*\sigma^2 + 6(L+r)^2\rho^2) \right. \\
& \left. + 14K\left(1-\alpha\right)^2 \eta_l^2 \left\| \nabla F(w_t) \right\| \right] + 28K^3\alpha^2(L+r)^4\eta_l^4\rho^2.
\end{aligned}
$$

*Proof.* The term $\mathbb{E} \left\| w_i^{t,k} - w^t \right\|^2$ can be rewriteen as

$$
\begin{aligned}
\mathbb{E} \left\| w_i^{t,k} - w^t \right\|^2 =& \mathbb{E} \left\| w_i^{t,k-1} - \eta_l \left[ \alpha \tilde{g}_i^{t,k-1} + (1 - \alpha\triangle^t) \right] - w^t \right\|^2 \\
\leq& \mathbb{E} \left\| w_i^{t,k-1} - w^t - \alpha\eta_l \left( \tilde{g}_i^{t,k-1} - \nabla F_i(\tilde{w}_i^{t,k-1}) + \nabla F_i(\tilde{w}_i^{t,k-1}) - \nabla F_i(\tilde{w}^t) + \nabla F_i(\tilde{w}^t) \right. \right. && \text{(11a)} \\
& \left. \left. - \nabla F(\tilde{w}^t) + \nabla F(\tilde{w}^t) \right) + (1 - \alpha)\eta_l \delta^t \right\|^2 \\
\leq& (1 + \frac{1}{2K-1} + 2\alpha^2(L+r)^2\eta_l^2)\mathbb{E} \left\| w_i^{t,k-1} - w^t \right\|^2 + 2\alpha^2(L+r)^2\eta_l^2\rho^2\sigma_l^2 && \text{(11b)} \\
& + 7K^2\alpha\eta_l^2\mathbb{E} \left\| \nabla F_i(\tilde{w}_i^{t,k-1}) - \nabla F_i(\tilde{w}) \right\|^2 + 7K\alpha^2\eta_l^2(3\sigma_g^2 + 6(L+r)^2\rho^2) \\
& + 7K\alpha^2\eta_l^2 \left\| \nabla F(\tilde{w}^t) \right\|^2 + 7K\eta_l^2(1-\alpha)^2 \left\| \triangle^t \right\|^2 \\
\leq& (1 + \frac{1}{2K-1} + 2\alpha^2(L+r)^2\eta_l^2 + 14K\alpha(L+r)^2\eta_l^2)\mathbb{E} \left\| w_i^{t,k-1} - w^t \right\|^2 + 2\alpha^2(L+r)^2\eta_l^2\rho^2\sigma_l^2 \\
& && \text{(11c)} \\
& + 7K(1-\alpha)^2\eta_l^2\mathbb{E} \left\| \triangle^t \right\|^2 + 14K\alpha^2(L+r)^2\eta_l^2\mathbb{E} \left\| \delta_i^{t,k} - \delta^t \right\|^2 \\
& + 7K\alpha^2\eta_l^2(3\sigma_g^2 + 6(L+r)^2\rho^2) + 7\alpha^2K\mathbb{E} \left\| \nabla F(\tilde{w}^t) \right\|^2 \\
\leq& (1 + \frac{1}{2K-1} + 2\alpha^2(L+r)^2\eta_l^2 + 14K\alpha(L+r)^2\eta_l^2)\mathbb{E} \left\| w_i^{t,k-1} - w^t \right\|^2 + 2\alpha^2(L+r)^2\eta_l^2\rho^2\sigma_l^2 \\
& && \text{(11d)} \\
& + 14K\alpha^2(L+r)^2\eta_l^2\mathbb{E} \left\| \delta_i^{t,k} - \delta^t \right\|^2 + 7K\alpha^2\eta_l^2(3\sigma_g^2 + 6(L+r)^2\rho^2) \\
& + 14K(1-\alpha)^2\eta_l^2 \left\| \nabla F(\tilde{w}^t) \right\|^2,
\end{aligned}
$$

$$
\text{(11e)}
$$

where (11a) follows from the fact that $\tilde{g}_i^{t,k-1}$ is an unbiased estimator of $\nabla F_i(\tilde{w}_i^{t,k-1})$ and Lemma 3; (11b) is from Lemma 2 and 5; (11c) is from Assumption 3 and Lemma 2; (11d) is from Assumption 2 and due to the fact that $\triangle \approx \nabla F(\tilde{w}^t)$ and $\alpha < \frac{1}{2}$.

Averaging over the clients $i$ and learning rate satisfies $\eta_l \leq \frac{1}{\sqrt{30}\alpha K(L+r)}$, we have:

$$\epsilon_{t,k} \leq (1 + \frac{1}{2K-1} + 2\alpha^2(L+r)^2\eta_l^2 + 14K\alpha(L+r)^2\eta_l^2)\mathbb{E}\left\|w_i^{t,k-1} - w^t\right\|^2$$

$$+ 2\alpha^2(L+r)^2\eta_l^2\rho^2\sigma_l^2 + 14K\alpha^2(L+r)^2\eta_l^2\mathbb{E}\left\|\delta_i^{t,k} - \delta^t\right\|^2$$

$$+ 7K\alpha^2\eta_l^2(3\sigma_g^2 + 6(L+r)^2\rho^2) + 14K(1-\alpha)^2\eta_l^2\left\|\nabla F(\tilde{w}^t)\right\|^2,$$

$$\leq (1 + \frac{1}{K-1})\frac{1}{N}\sum_{i=1}^{N}\mathbb{E}\left\|w_i^{t,k-1} - w^t\right\|^2 \tag{12a}$$

$$+ 2\alpha^2(L+r)^2\eta_l^2\rho^2\sigma_l^2 + 14K\alpha^2(L+r)^2\eta_l^2\mathbb{E}\left\|\delta_i^{t,k} - \delta^t\right\|^2$$

$$+ 7K\alpha^2\eta_l^2(3\sigma_g^2 + 6(L+r)^2\rho^2) + 14K(1-\alpha)^2\eta_l^2\left\|\nabla F(\tilde{w}^t)\right\|^2,$$

$$\leq \sum_{\tau=0}^{k-1}(1 + \frac{1}{K-1})^{\tau}\left[2\alpha^2(L+r)^2\eta_l^2\rho^2\sigma_l^2 + 7K\alpha^2\eta_l^2(3\sigma_g^2 + 6(L+r)^2\rho^2)\right.$$

$$\left. + 14K\alpha^2(L+r)^2\eta_l^2\mathbb{E}\left\|\delta_i^{t,k} - \delta^t\right\|^2\right] + 14K(1-\alpha)^2\eta_l^2\left\|\nabla F(\tilde{w}^t)\right\|^2,$$

$$\leq 5K\left(2\alpha^2(L+r)^2\eta_l^2\rho^2\sigma_l^2 + 7K\alpha^2\eta_l^2(3\sigma_g^2 + 6(L+r)^2\rho^2) + 14K(1-\alpha)^2\eta_l^2\left\|\nabla F(\tilde{w}^t)\right\|^2\right) \tag{12b}$$

$$+ 28\alpha^2 K^3(L+r)^4\eta_l^4\rho^2,$$

where (12a) is due to the fact that $\eta_l \leq \frac{1}{\sqrt{30}\alpha K(L+r)}$ and $\alpha \leq \frac{1}{2}$; (12b) is from Lemma B.1 in [31]. $\qquad\square$

The global update can be bounded by

**Lemma 7.** *For the partial client participation, we can bound $\mathbb{E}_t\left[\|\triangle^t\|^2\right]$ as follows:*

$$\mathbb{E}_t\left[\left\|\triangle^{t+1}\right\|^2\right] \leq \frac{K\eta_l^2\rho^2\sigma_l^2}{S}(L+r)^2 + \frac{\eta_l^2}{S^2}\left[\left\|\sum_{i=1}^{N}\mathbb{P}\left\{i \in S^t\right\}\sum_{k=0}^{K-1}\nabla F_i(\tilde{w}_i^{t,k})\right\|^2\right]$$

*Proof.*

$$\mathbb{E}_t\left[\left\|\triangle^{t+1}\right\|^2\right] = \frac{1}{K^2S^2\eta_l^2}\mathbb{E}_t\left[\sum_{i\in S_t}\left\|\sum_k\left(\alpha\eta_l\tilde{g}_i^{t,k} + \eta_l(1-\alpha)\triangle^t\right)\right\|^2\right]$$

$$= \frac{\alpha^2}{K^2S^2}\mathbb{E}_t\left[\sum_{i\in S_t}\left\|\sum_{k=0}^{K-1}\tilde{g}_i^{t,k} - \nabla F_i(\tilde{x}_i^{t,k})\right\|^2\right] + \frac{1}{K^2S^2}\mathbb{E}_t\left[\sum_{i\in S_t}\left\|\sum_k\left(\alpha\nabla F_i(\tilde{x}_i^{t,k}) + (1-\alpha)\triangle^t\right)\right\|^2\right] \tag{13a}$$

$$\leq \frac{\alpha^2(L+r)^2\rho^2}{KS}\sigma_l^2 + \frac{2(1-\alpha^2)}{KS}\left\|\nabla F(\tilde{w}^t)\right\|^2 + \frac{2\alpha^2}{K^2S^2}\left[\sum_i\mathbb{P}\left\{i \in S_t\right\}\left\|\sum_{k=0}^{K-1}\nabla F_i(w_i^{t,k})\right\|^2\right] \tag{13b}$$

$$= \frac{\alpha^2(L+r)^2\rho^2}{KS}\sigma_l^2 + \frac{2(1-\alpha^2)}{KS}\left\|\nabla F(\tilde{w}^t)\right\|^2 + \frac{2\alpha^2}{K^2SN}\sum_{i=1}^{N}\mathbb{E}_t\left\|\sum_{k=0}^{K-1}\nabla F_i(\tilde{w}_i^{t,k})\right\|^2$$

$$+ \frac{2\alpha^2(S-1)}{K^2SN^2}\mathbb{E}_t\left\|\sum_{i=1}^{N}\sum_{k=0}^{K-1}\nabla F_i(\tilde{w}_i^{t,k})\right\|^2, \tag{13c}$$

where (13a) is from Lemma 5 and (13b) is from Lemma 4. □

Next, we provide the following lemma to demonstrate the descent behavior of FedMRUR under partial client participation setting.

**Lemma 8.** *For all $t \in [T-1]$ and $i \in S_t$, with the choice of learning rate, the iterates generated by FedMRUR under partial client participation satisfy:*

$$
\begin{aligned}
\mathbb{E}_t\left[F(w^{t+1})\right] \leq & F(\tilde{w}^t) - K\eta_g\eta_l d_t(\frac{1}{2} - 20K^2L^2\eta_l^2)\left\|\nabla F(\tilde{w}^t)\right\|^2 + K\eta_g\eta_l\left(6K^2\eta_l^2\alpha^4\rho^2\right.\\
& \left. + 5K^2\eta_l\alpha^4\rho^2\sigma^2 + 20K^3\eta_l^3\alpha^2\sigma_g^2 + 16K^3\eta_l^4\alpha^6\rho^2 + \frac{\eta_g\eta_l\alpha^3\rho^2}{N}\sigma_l^2\right).
\end{aligned}
$$

*Proof.* Let's define $\epsilon_\delta = \frac{1}{N}\sum_i \mathbb{E}\left[\delta_{i,k} - \delta\right]^2$, where $\delta = \underset{\delta}{\arg\max}\, F(w + \delta)$.

$$
\begin{aligned}
\mathbb{E}_t\left[F(w^{t+1})\right] \leq & F(w^t) + E_t\left\langle\nabla F(\tilde{w}^t), \tilde{w}^{t+1} - \tilde{w}^t\right\rangle + \frac{L+r}{2}\mathbb{E}_t\left[\left\|\tilde{w}^{t+1} - \tilde{w}^t\right\|^2\right]\\
= & F(w^t) - \alpha\eta_g\left\|\nabla F(\tilde{w}^t)\right\|^2 + \eta_g\left\langle\nabla F(\tilde{w}^t), \mathbb{E}\left[-\triangle^{t+1} + \alpha\nabla F(\tilde{w}^t)\right]\right\rangle + \frac{L+r}{2}\eta_g^2\mathbb{E}_t\left[\left\|\triangle^{r+1}\right\|^2\right]
\end{aligned}
$$
(14)

Let's denote the $\frac{\sum_{i \in S_t}\|\triangle_i^t\|}{\|\sum_{i \in S_t}\triangle_i^t\|}$ as $d_t$ and bound the third term in (14) as follows:

$$
\begin{aligned}
\left\langle\nabla F(\tilde{w}^t), \mathbb{E}\left[-\triangle^{t+1} + \alpha\nabla F(\tilde{w}^t)\right]\right\rangle \leq & \left(\frac{3\alpha}{2} - 1\right)d_t\left\|\nabla F(\tilde{w}^t)\right\|^2 + \alpha(L+r)^2 d_t(\epsilon_{t,k} + \epsilon_\delta)\\
& - \frac{\alpha d_t}{2K^2N^2}\mathbb{E}_t\left\|\sum_{i,k}\nabla F_i(\tilde{w}_i^{t,k})\right\|^2
\end{aligned}
$$
(15)

Plugging (15) into (14), we have:

$$
\mathbb{E}_t\left[F(\tilde{w}^{t+1})\right]
$$

$$
\leq F(\tilde{w}^t) - \left(\eta_g - \frac{\alpha\eta_g}{2}\right)d_t\left\|\nabla F(\tilde{w}^t)\right\|^2 + \alpha(L+r)^2\eta_g d_t(\epsilon_{t,k} + \epsilon_\delta)
$$

$$
- \frac{\alpha\eta_g d_t}{2K^2N^2}\mathbb{E}_t\left\|\sum_{i,k}\nabla F_i(\tilde{w}_i^{t,k})\right\|^2 + \frac{(L+r)\eta_g^2}{2}\mathbb{E}_t\left[\left\|\triangle^{t+1}\right\|^2\right]
$$

$$
\leq F(\tilde{w}^t) - \left(\frac{3\alpha\eta_g d_t}{4} - \frac{2(1-\alpha)^2(L+r)\eta_g d_t}{KS}\right)\left\|\nabla F(\tilde{w}^t)\right\|^2 + \alpha(L+r)^2\eta_g d_t(\epsilon_{t,k} + \epsilon_\delta)
$$

(16a)

$$
+ \frac{\alpha^2(L+r)^3\rho^2\eta_g^2}{2KS}\sigma_l^2 - \frac{\alpha\eta_g d_t}{2K^2N^2}\mathbb{E}_t\left\|\sum_{i,k}\alpha\nabla F_i(\tilde{w}_i^{t,k})\right\|^2
$$

$$
+ \frac{(L+r)\alpha^2\eta_g^2}{2K^2SN}\sum_i\mathbb{E}_t\left\|\sum_k\nabla F_i(\tilde{w}_i^{t,k})\right\|^2 + \frac{(L+r)\alpha^2(S-1)\eta_g^2}{K^2SN^2}\mathbb{E}_t\left\|\sum_k\nabla F_i(\tilde{w}_i^{t,k})\right\|^2
$$

$$
\leq F(\tilde{w}^t) - \alpha\eta_g d_t\left(\frac{3}{4} - \frac{2(1-\alpha)(L+r)}{KN} - 70(1-\alpha)K^2(L+r)^2\eta_l^2 - \frac{90\alpha(L+r)^3\eta_g\eta_l^2}{Sd_t}\right.
$$

(16b)

$$
\left. - \frac{3\alpha(L+r)\eta_g}{2S}\right)\left\|\nabla F(\tilde{w}^t)\right\|^2 + \beta\eta_g\left(10\alpha^2(L+r)^4\eta_l^2\rho^2\sigma_l^2 + 28\alpha^2K^3(L+r)^6\eta_l^4\rho^2\right.
$$

$$
+ 35\alpha^2K(L+r)^2\eta_l^2(3\sigma_g^2 + 6(L+r)^2\rho^2) + 2K^2(L+r)^4\eta_l^2\rho^2 + \frac{\alpha(L+r)^3\eta_g^2d_t^2\rho^2}{2KS}\sigma_l^2
$$

$$
+ \frac{\alpha L\eta_g d_t}{K^2SN}(30NK^2L^4\eta_l^2\rho^2\sigma_l^2 + 270NK^3(L+r)^2\eta_l^2\sigma_g^2 + 540NK^2(L+r)^4\eta_l^2\rho^2
$$

$$
+ 72K^4(L+r)^6\eta_l^4\rho^2 + 6NK^4L^2\eta_l^2\rho^2 + 4NK^2\sigma_g^2 + 3NK^2(L+r)^2\rho^2))
$$

$$
\leq F(\tilde{w}^t) - C\alpha\eta_g d_t\left\|\nabla F(\tilde{w}^t)\right\|^2 + \beta\eta_g\left(10\alpha^2(L+r)^4\eta_l^2\rho^2\sigma_l^2 + 28\alpha^2K^3(L+r)^6\eta_l^4\rho^2\right. \quad \text{(16c)}
$$

$$
+ 35\alpha^2K(L+r)^2\eta_l^2(3\sigma_g^2 + 6(L+r)^2\rho^2) + 2K^2(L+r)^4\eta_l^2\rho^2 + \frac{\alpha(L+r)^3\eta_g^2d_t^2\rho^2}{2KS}\sigma_l^2
$$

$$
+ \frac{\alpha L\eta_g d_t}{K^2SN}(30NK^2L^4\eta_l^2\rho^2\sigma_l^2 + 270NK^3(L+r)^2\eta_l^2\sigma_g^2 + 540NK^2(L+r)^4\eta_l^2\rho^2
$$

$$
+ 72K^4(L+r)^6\eta_l^4\rho^2 + 6NK^4L^2\eta_l^2\rho^2 + 4NK^2\sigma_g^2 + 3NK^2(L+r)^2\rho^2))
$$

(16d)

where (16a) is from Lemma 7; (16b) is from Lemmas 6, Lemma B.1 in [31] and due to the fact that $\eta_g \leq \frac{S}{2\alpha L(S-1)}$ and (16c) is from the condition that $\frac{3}{4} - \frac{2(1-\alpha)L}{KN} - 70(1-\alpha)K^2(L+r)^2\eta_l^2 - \frac{90\alpha(L+r)^3\eta_g\eta_l^2}{S} - \frac{3\alpha(L+r)\eta_g}{2S} > C > 0$ and $\alpha \leq \frac{1}{2}$ hold. $\qquad\square$

Finally, we provide following two theorems to charcterize the convergence rate of FedMRUR:

**Theorem 9** (Extension of Theorem 1). *Let all the assumptions hold and with partial client participation. If we choose learning rate $\eta_l \leq \frac{1}{\sqrt{30}\alpha KL}$, $\eta_g \leq \frac{S}{2\alpha L(S-1)}$ satisfying $\frac{3}{4} - \frac{2(1-\alpha L)}{KN} - 70(1 - \alpha)K^2(L+r)^2\eta_l^2 - \frac{90\alpha(L+r)^3\eta_g\eta_l^2}{S} - \frac{3\alpha(L+r)\eta_g}{2S}$, then for all $K \geq 0$ and $T \geq 1$, we have:*

$$
\frac{1}{\sum_{t=1}^T d_t}\sum_{t=1}^T\mathbb{E}\left\|\nabla F(w^t)\right\|^2 d_t \leq \frac{F^0 - F^*}{C\alpha\eta_g\sum_{t=1}^T d_t} + \Phi,
$$

*where*

$$\Phi = \frac{1}{C}\Big[10\alpha^2(L+r)^4\eta_l^2\rho^2\sigma_l^2 + 35\alpha^2K(L+r)^2\eta_l^2 3(\sigma_g^2 + 6(L+r)^2\rho^2) + 28\alpha^2K^3(L+r)^6\eta_l^4\rho^2$$

$$+ 2K^2L^4\eta_l^2\rho^2 + \frac{\alpha(L+r)^3\eta_g^2\rho^2}{2KS}\sigma_l^2 + \frac{\alpha(L+r)\eta_g}{K^2SN}(30NK^2(L+r)^4\eta_l^2\rho^2\sigma_l^2$$

$$+ 270NK^3(L+r)^2\eta_l^2\sigma_g^2 + 540NK^2(L+r)^4\eta_l^2\rho^2 + 72K^4(L+r)^6\eta_l^4\rho^2$$

$$+ 6NK^4(L+r)^2\eta_l^2\rho^2 + 4NK^2\sigma_g^2 + 3NK^2(L+r)^2\rho^2)\Big].$$

*If we set $\eta_g = \Theta(\frac{\sqrt{SK}}{\sqrt{T}})$ and $\eta_l = \Theta\left(\frac{1}{\sqrt{ST}K(L+r)}\right)$, the convergence rate of the FedMRUR under partial client participation is:*

$$\frac{1}{T}\sum_{t=1}^{T}\mathbb{E}\left\|\nabla F(w^t)\right\|^2 = O(\frac{1}{\sqrt{SKT}}) + O\left(\frac{\sqrt{K}}{ST}\right) + O\left(\frac{1}{\sqrt{K}T}\right).$$

*Proof.* Summing (16c) in Lemma 8 over $t = \{1, ..., T\}$ and multiplying both sides by $\frac{1}{C\alpha\eta_g\sum_{t=1}^{T}d_t}$, we have

$$\frac{1}{T}\sum_{t=1}^{T}\mathbb{E}\left\|\nabla F(w^t)\right\|^2 \leq \frac{F(\tilde{w}^t - F(\tilde{w}^{t+1}))}{C\alpha\eta_g\sum_{t=1}^{T}d_t} + \Phi$$

$$\leq \frac{F^0 - F^*}{C\alpha\eta_g\sum_{t=1}^{T}d_t} + \Phi,$$

where the second inequality comes from the fact that $F^0 - F^* \leq F(\tilde{w}^t) - F(\tilde{w}^{t+1})$. According to the definition of $d_t$ in Lemma 8 and triangle inequality, we have $1 \leq \frac{\sum_{i\in S_t}\|\triangle_i^t\|}{\|\sum_{i\in S_t}\triangle_i^t\|} \leq S$ and $\sum_{t=1}^{T}d_t \geq T$. If we choose $\eta_g = \Theta(\frac{\sqrt{SK}}{\sqrt{T}})$, $\eta_l = \Theta\left(\frac{1}{\sqrt{ST}K(L+r)}\right)$ and $\rho = \Theta(\frac{1}{\sqrt{T}})$, the above ineqaulity can be rewriteen as

$$\frac{1}{T}\sum_{t=1}^{T}\mathbb{E}\left\|\nabla F(w^t)\right\|^2 = O(\frac{1}{\sqrt{SKT}}) + O\left(\frac{\sqrt{K}}{ST}\right) + O\left(\frac{1}{\sqrt{K}T}\right).$$

$\square$

## B Experiments

### B.1 Results for CIFAR-10

Table 5 characterizes the convergence speed of multiple algorithms on CIFAR-10. For most of the time, our proposed method, FedMRUR outperforms the baselines. Therefore, we can conclude that: 1) our method achieves the fastest convergence speed, especially when the data heterogeneity is large, which validates the normalized update aggregation scheme accelerate the iteration; 2) when the statistical heterogeneity is large, the proposed FedMRUR accelerates the convergence more effectively, which validates that utilizing the hyperbolic graph fusion is able to alleviate the issue of the model inconsistency across clients.

Table 6 presents the final test accuracy of ResNet-18 trained using multiple algorithms on CIFAR-10 dataset under four heterogeneous settings. We plot the test accuracy of the algorithms for the image classification task in Figure 5. We can oberserve that the proposed FedMRUR performs well with good stability and efficently mitigates the negative effect of the model inconsistency. Specifically, on the Dirichlet-0.3 setups, FedMRUR achieves a test accuracy of 84.53%, which is 0.51% higher than the second-best algorithm, MoFedSAM. Based on these, we can conclude that FedMRUR reduces the model inconsistency and improves the convergence speed effectively.

### B.2 Verification of Normalized Aggregation

From the theoretical view, we can conclude that the "Normalized Aggregation of Local Updates" can accelerate the convergence in Theorem 9. In fact, using this operator in other baselines can also

Table 5: Convergence speed on CIFAR-10 dataset in both Dir($\mu$) and Path($n$) distributions. "Acc." represents the target test accuracy on the dataset. "$\infty$" means that the algorithm is unable to achieve the target accuracy on CIFAR-10 dataset.

| Algorithms | | FedAvg | FedExp | FedProx | SCAFFOLD | FedCM | MoFedSAM | Our |
|---|---|---|---|---|---|---|---|---|
| ACC. | | | | | 70% | | | |
| Dir($\mu$) | 0.6 | 392 | 538 | 354 | 263 | 95 | 119 | 125 |
| | 0.3 | 513 | 518 | 452 | 349 | 131 | 134 | 142 |
| Path($n$) | 6 | 353 | 459 | 328 | 242 | 110 | 112 | 115 |
| | 3 | $\infty$ | 770 | $\infty$ | 466 | 177 | 178 | 192 |
| ACC. | | | | | 75% | | | |
| Dir($\mu$) | 0.6 | $\infty$ | 788 | $\infty$ | 441 | 185 | 178 | 180 |
| | 0.3 | $\infty$ | 905 | $\infty$ | 588 | 229 | 205 | 221 |
| Path($n$) | 6 | $\infty$ | 866 | $\infty$ | 426 | 171 | 166 | 166 |
| | 3 | $\infty$ | 1225 | $\infty$ | 1552 | 278 | 307 | 305 |
| ACC. | | | | | 80% | | | |
| Dir($\mu$) | 0.6 | $\infty$ | $\infty$ | $\infty$ | $\infty$ | 471 | 384 | 393 |
| | 0.3 | $\infty$ | $\infty$ | $\infty$ | $\infty$ | 573 | 450 | 449 |
| Path($n$) | 6 | $\infty$ | $\infty$ | $\infty$ | 1181 | 443 | 356 | 353 |
| | 3 | $\infty$ | $\infty$ | $\infty$ | $\infty$ | 810 | 636 | 630 |

Table 6: Test accuracy (%) on CIFAR-10 dataset in both Dir($\mu$) and Path($n$)) distributions.

| Algorithm | CIFAR-10 | | | |
|---|---|---|---|---|
| | Dir($\mu$) | | Path($n$) | |
| | $\mu = 0.6$ | $\mu = 0.3$ | $n = 6$ | $n = 3$ |
| FedAvg | 72.96 | 71.44 | 73.22 | 67.78 |
| FedExp | 79.26 | 76.73 | 78.70 | 74.65 |
| FedProx | 73.69 | 72.15 | 73.95 | 68.46 |
| SCAFFOLD | 79.69 | 78.49 | 79.77 | 72.72 |
| FedCM | 84.48 | 82.95 | 84.15 | 83.10 |
| MoFedSAM | 84.99 | 84.03 | 85.10 | 84.13 |
| FedMRUR | 85.70 | 84.53 | 85.61 | 84.89 |

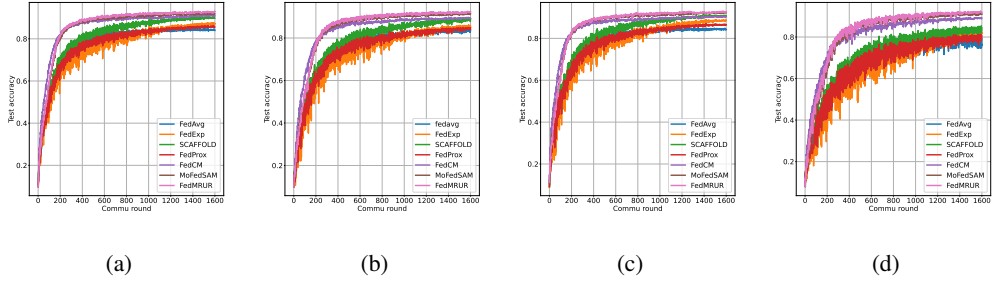

(a)     (b)     (c)     (d)

Figure 5: Test accuracy w.r.t. communication rounds of our proposed method and other approaches. Each method performs in 1600 communication rounds. To compare them fairly, the basic optimizers are trained with the same hyperparameters on CIFAR-10 dataset.

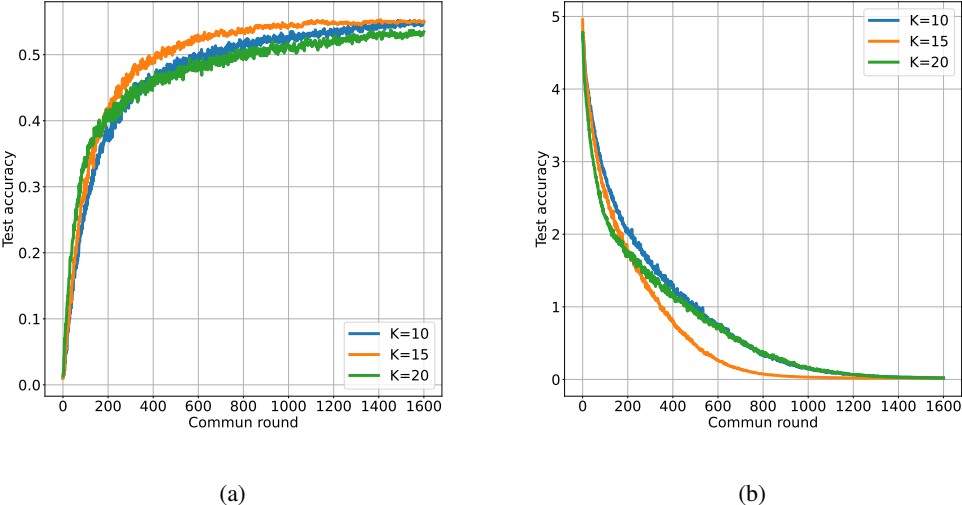

(a)                                            (b)

Figure 6: Test accuracy and train loss w.r.t communication rounds of FedMRUR with different local intervals $K$.

improve the performance. Here, we present the effect of the normalized aggregation method applied to FedCM and FedAvg in Table 7. From the results, we can find that the "Normalized Aggregation" can improve the convergence a lot (For example, when $\mu = 0.3$, it can improve the final acc $4\%$ over FedCM).

Table 7: Test accuracy (%) on CIFAR-10 dataset in both Dir($\mu$) and Path($n$)) distributions.

| Algorithm | CIFAR-100 | | | |
|---|---|---|---|---|
| | Dir($\mu$) | | Path($n$) | |
| | $\mu = 0.6$ | $\mu = 0.3$ | n = 20 | n = 10 |
| $FedAvg$ | 39.87 | 39.50 | 38.47 | 36.67 |
| $FedAvg^+$ | 42.09 | 41.71 | 42.22 | 40.10 |
| $FedCM$ | 51.01 | 50.93 | 50.58 | 50.03 |
| $FedCM^+$ | 52.53 | 52.32 | 52.59 | 52.50 |

### B.3 Validation for the linear speedup propoerty

In this part, we present the experiment results which veritfies the linear speedup propoerty of the proposed FedMRUR. Because the whole dataset is fixed, increasing the number of clients changes the amount of data in the local data, which changes the entire optimization problem, we conduct the experiment under various settings of local intervals $K$ fixing the number of clients to verify the linear speedup property. From Figure 6, when $K$ increase to 15, the algorithm achieves $1.5\times$ than $K = 10$. From (8), when local interval $K$ is increased to $\mathcal{O}\left(ST\right)^{\frac{1}{2}}$, the impact of the second term in Theorem 1 becomes greater and the first term becomes less. Therefore, when $K$ increase from 15 to 20, the speedup of convergence is not obvious.

### B.4 Impact of Hyperbolic space

Since hyperbolic geometry is a Riemann manifold with a constant negative curvature, its typical geometric property is that the volume grows exponentially with its radius, whereas the Euclidean space grows polynomially. Such a geometric trait has 2 advantages:

- The hyperbolic space exhibits minimal distortion and it fits the hierarchies particularly well since the space closely matches the growth rate of graph-like data while the Euclidean space doesn't.
- Even with a low-embedding dimensional space, hyperbolic models are surprisingly able to produce a high quality representation, which makes them to be particularly advantageous in low-memory and low-storage scenarios.

In realistic scenarios, there exists many graph-like data structure, such as the hypernym structure in NLP, the subordinate structure of entities in the knowledge graph and the power-law distribution in recommender systems. In FL, the machine learning models have a graph-like structure, so adopting the Lorentz metric of hyperbolic space makes use of the hierarchical information in neural networks, which are helpful to fuse the model further bring prediction gains. Using Euclidean metric, or some Riemann metric defined by a positive definite matrix is an interesting idea. Here, we show the results of experiments using different geometric spaces as follow (Table 8). From these results, we can find that Lortenz metric of hyperbolic space can help the algorithm achieving the highest test accuracy.

Table 8: Test accuracy (%) on CIFAR-100 dataset using different manifolds.

| Space | Euclidean | Hyperbolic |
|---|---|---|
| Test Acc. | 54.01(0.36) | 55.64(0.41) |

The representations generated by the model have fewer dimensions than the data. Mapping the representations to the hyperbolic space introduces less computation overhead than mapping the data. We also conduct experiments mapping the original data to the hyperbolic space over 8 seeds. The results are as presented in Table 9.

Table 9: Test accuracy (%) on CIFAR-100 dataset using hyperbolic model in different ways.

| Orignial data | Representation |
|---|---|
| 56.03(0.56) | 55.64(0.41) |

From the table, we can see that both methods achieve similar performance. Thus, considering the computation overhead and performance, we only map representations to hyperbolic space and do not treat the entire learning process in hyperbolic space.

To study the impact of $\beta$ for hyperbolic graph manifold regularization on the performance, we conduct the experiment on CIFAR100 task with different $\beta$ settings and present the final test accuracy in Table 10. From this table, we can find that has a limited impact on the final performance of the algorithm.

Table 10: Test accuracy (%) on CIFAR-100 dataset using different $\beta$.

| $\beta$ | 0.1 | 0.5 | 1.0 | 5.0 | 10.0 |
|---|---|---|---|---|---|
| Test Acc. | 54.67 | 54.69 | 55.04 | 54.79 | 54.91 |

## B.5 Training time

**Test Experiments:** Nvidia GTX-3090 GPU, CUDA Driver 11.4, Driver Version 470.10.3.01, Pytorch-1.11.1

Table 11 shows the wall-clock time costs on the CIFAR-100 of Dirichlet-0.3 dataset split. Due to the double computation of the gradients via SAM optimizer, MoFedSAM and FedMRUR will take more time in a single communication round, about $1.46\times$ over the FedCM method. However, the communication rounds required is less than FedCM. Considering the total wall-clock time costs, the acceleration ratio of FedMRUR achieves $2.46\times$ compared with MoFedSAM ($3.67\times$ compared with FedCM) at the final. Therefore, we can conclude that the FedMRUR is more efficient with respect to the communication round and wall-clock time when high-performance models are required.

Table 11: Test accuracy (%) on CIFAR-100 dataset to achieve 50% test accuracy.

| Algorithm | Times(s/round) | Rounds | Total(s) | Cost Ratio |
|-----------|----------------|--------|----------|------------|
| FedAvg | 9.23 | $\infty$ | $\infty$ | $\infty$ |
| FedExp | 14.82 | $\infty$ | $\infty$ | $\infty$ |
| SCAFFOLD | 15.52 | $\infty$ | $\infty$ | $\infty$ |
| FedProx | 12.89 | $\infty$ | $\infty$ | $\infty$ |
| FedCM | 11.53 | 1407 | 16222.71 | $3.67\times$ |
| MoFedSAM | 15.53 | 701 | 10886.53 | $2.46\times$ |
| FedMRUR | 16.82 | 263 | 4423.66 | $1\times$ |