# OpenReview forum: "Federated Learning with Manifold Regularization and Normalized Update Reaggregation"
_NeurIPS.cc/2023/Conference — NeurIPS 2023 poster_

### Official Review · Reviewer_EAwW · 2023-07-04

**Soundness:** 3 good
**Presentation:** 1 poor
**Contribution:** 3 good
**Rating:** 5
**Confidence:** 4

**Summary:**

The authors proposed a new algorithm for federated learning based on a Lorentzian regularization. The proposed algorithm achieves desired sub-linear convergence with linear speed-up. Numerical experiment shows the efficacy of the proposed algorithm over existing federated learning algorithms

**Strengths:**

The proposed method is theoretically well-rounded with convergence guarantee for smooth non convex situation, which is the most commonly setting in the literature.

**Weaknesses:**

(Please respond to the Questions section directly) The presentation of the paper is confusing. Also some technical details are not well-illustrated. The theory part of the paper is not clearly presented and seems to be abusing terminologies a lot.

**Questions:**

Major:

1. The presentation of the paper is confusing: First, I cannot link the proposed Algorithm 1 with all the argument in Section 3.2. More specifically, how is the “mapped Lorentzian vectors” in (3) used in Algorithm 1? The authors claimed that the regularized problem (5) has a number of desired property but is Algorithm 1 solving this regularized problem? The argument in this section seems to be ending at nowhere; Second, I really cannot understand the usages of “hyperbolic” and “Lorentzian”. The hyperbolic space is defined as space with constant -1 curvature in mathematics, but here we are not dealing with curvatures at all. As for the Lorentzian regularizer defined in (4) and (5) (which again I don’t understand how it is used in Algorithm 1), why do we need Lorentz metric? What would happen if we just use Euclidean metric, or some Riemannian metric defined by a positive definite matrix? The entire section 3.2 is confusing. On the other hand, all the described operations in Section 3.2 are not well-reflected in Algorithm 1. For example, what’s $g_{i}^{t,k}$ in Algorithm 1?

2. As the authors claimed, the algorithm is built upon MoFedSAM[1]. How is the rate of convergence (in theory) of Algorithm 1 compared with existing works, especially MoFedSAM? If the theoretical rate is not improved, I would argue that this refined engineering over MoFedSAM might not be very interesting since (in my perspective) the only difference of Algorithm 1 with MoFedSAM is the $\Delta_i$ which serve as a local update corrector.

3. I cannot understand what point Figure 2 try to make. To me Figure 2(b) is a rotation operation, not projection, also I don’t understand how is the operation used in Algorithm 1.


Minor:
1. In the abstract, the authors include a formula. I think it’s better to either explain the meaning of each of the variable in the formula, or just use words instead;

2. What does the name “FedMRUR” exactly stand for?

3. Typos: Line 227, Assumption 1, missed one $\nabla$; Line 263 “Dirichlet… from {0.30.6}” missed a comma?


References:
[1] Qu, Zhe, et al. "Generalized federated learning via sharpness aware minimization." International Conference on Machine Learning. PMLR, 2022.

**Limitations:**

The limitation is well stated in weakness and question sections.

I’m not aware of any potential negative social impact of this work.

---

> ### Author Rebuttal · Authors · 2023-08-09
>
> Thank you for the constructive comments which helps us improve the paper. We have prepared our responses to each of your questions.
>
> **- Q1(major):The presentation of the paper is confusing:
>     - I cannot link the proposed Algorithm 1 with all the argument in Section 3.2. More specifically, how is the “mapped Lorentzian vectors” in (3) used in Algorithm 1? The authors claimed that the regularized problem (5) has a number of desired property but is Algorithm 1 solving this regularized problem? The argument in this section seems to be ending at nowhere;
>     - I really cannot understand the usages of “hyperbolic” and “Lorentzian”. The hyperbolic space is defined as space with constant -1 curvature in mathematics, but here we are not dealing with curvatures at all. As for the Lorentzian regularizer defined in (4) and (5) (which again I don’t understand how it is used in Algorithm 1), why do we need Lorentz metric? What would happen if we just use Euclidean metric, or some Riemann metric defined by a positive definite matrix? The entire section 3.2 is confusing.
>     - On the other hand, all the described operations in Section 3.2 are not well-reflected in Algorithm 1. For example, what’s $g_{i}^{t,k}$ in Algorithm 1?**
> - A1(major): For the answer, please refer to Q1 in the Author Rebuttal.
>
> **- Q2(major): As the authors claimed, the algorithm is built upon MoFedSAM. How is the rate of convergence (in theory) of Algorithm 1 compared with existing works, especially MoFedSAM? If the theoretical rate is not improved, I would argue that this refined engineering over MoFedSAM might not be very interesting since (in my perspective) the only difference of Algorithm 1 with MoFedSAM is the $\triangle_i$ which serve as a local update corrector.**
> - A2(major): For the answer, please refer to Q2 in  the Author Rebuttal.
>
> **- Q3(major): I cannot understand what point Figure 2 try to make. To me Figure 2(b) is a rotation operation, not projection, also I don’t understand how is the operation used in Algorithm 1.**
> - A3(major): Sorry for the unclear presentation of Figure 2. Figure 2 is a toy schema to compare the naive aggregation and normalized aggregation of the local updates. The main difference is the norm $\lVert \triangle \rVert$ of the global update, which is the aggregation of the local updates $\triangle_p$ from the participating clients. When adopting the naive aggregation method, the server directly takes the average of the local updates as the global update. Each client's contribution on the global update is $\lVert \triangle_p \rVert \cos \theta_p$, where $\theta_p$ is the angle between the local update $\triangle_p$ and the global update $\triangle$. When the data heterogeneity is significant, the divergence between the directions of local updates is large. Correspondingly, the divergence between the directions of each local update and the global update is large, and the global update norm shrinks which slows down the convergence of the global objective function. When adopting the normalized aggregation method, the server computes the direction of the global update by normalizing the sum of local updates, which is the same as the naive method. The global update norm is obtained by averaging the local updates. The normalized aggregation way is depicted in line 14 of Algorithm 1. In this way, the client $p$'s contribution on the global update $\theta$ grows from $\lVert \triangle_p \rVert \cos \theta_p$ to $\lVert \triangle_p \rVert$. Accordingly, the norm of the global update grows and the speedup the convergence. In order to compare the two aggregation methods, we repalce Figure 2 with a new one (Figure 2 in the attached pdf).
>
> **- Q1(minor): In the abstract, the authors include a formula. I think it’s better to either explain the meaning of each of the variable in the formula, or just use words instead;**
> A1(minor): Thank you for the advice about the abstract. We explain the meaning of variables $S$ (the number of participated clients in one communication round), $K$ (the local interval), $T$(the total number of communication rounds) in the formula.
>
> **- Q2(minor): What does the name “FedMRUR” exactly stand for?**
> A2(minor): Thanks for the reminder. In the revision of the paper, we include the full name where FedMRUR first appears. The name "FedMRUR" stands for **Fed**erated Learning with **M**anifold **R**egularization and Normalized **U**pdate **R**eaggregation. In the name "FedMRUR", we want to emphasize two main components in our algorithm: the manifold model fusion scheme using the hyperbolic model to constrain the divergence between the local and global model and the normalized client updates aggregation way applied at the server to mitigate the norm reduction of the global update.
>
> **- Q3(minor): Typos: Line 227, Assumption 1, missed one $\nabla$; Line 263 “Dirichlet… from {0.30.6}” missed a comma?**
> A3(minor): Thank you for the careful proofreading. We add the missed $\nabla$ in the Assumption 1 at Line 215 (the new draft). At Line 252, we add the missed comma and the sentence is "Dirichlet coefficient $\mu$ from {0.3 , 0.6}".

---

> > ### Comment · Reviewer_EAwW · 2023-08-14
> >
> > I thank the authors for responses to my questions. May I know where I can find the revised version of the paper? For example I cannot find Theorem 9 mentioned in the rebuttal.
> >
> > I still have some concerns with this work:
> >
> > The first issue is that adding the extra hyperbolic regularization may increase the testing accuracy, but the justification in the author's rebuttal "In FL, clients and the server logically also have a tree-like hierarchical topological relationship, so adopting the Lorentz metric of hyperbolic space makes use of the hierarchical information in the datasets and the hierarchical relationship between the server and clients, which are helpful to group data samples further bring prediction gains" seems too strong and general to be just supported by current evidences. However this is still not the biggest issue that I'm concerned, since the observation may be a good starting point for incorporating geometric information in FL. The authors could give a comprehensive study on the effectiveness of different regularization terms as a future work to support their broad claim.
> >
> > My biggest concern is that I'm still not sure if the proposed algorithm is theoretically better than MoFedSAM[1]. As in the rebuttal "The main dominated term of convergence rate is ..., the convergence rate of FedMRUR is faster than MoFedSAM". Can the authors illustrate why the rate $(F^0-F^*)/(c\alpha\eta_g \sum_{t=1}^{T}d_t) + \Phi$ is better than $(F^0-F^*)/(c\alpha\eta_g T)$ in [1], as the authors wrote in the rebuttal? I'm still very concerned and confused here.
> >
> > [1] Qu, Zhe, et al. "Generalized federated learning via sharpness aware minimization." International Conference on Machine Learning. PMLR, 2022.

---

> > > ### Author Response · Authors · 2023-08-15
> > >
> > > **-Q1: May I know where I can find the revised version of the paper? Where is Theorem 9.**
> > >
> > > -A1: We are sorry about that you can't find the revised paper. Due to the limitations of the rebuttal system, we can't upload the revised paper. We re-analyze the convergence rate of the algorithm to better illustrate the effect of **normalized aggregation**. Theorem 9 is the revised version of Theorem 8 in the supplementary file. Compared with Theorem 8, under the same condition, Theorem 9 gives a tighter upper bound on the convergence rate, which is
> > >     $$\frac{1}{\sum_{t=1}^{T} d_t} \sum_{t=1}^{T} \mathbb{E} \left\lVert \nabla F(w^t) \right\rVert^2 d_t \le \frac{F^0 - F^*}{C \alpha \eta_g \sum_{t=1}^{T} d_t} + \Phi.$$
> > > In A2(the reply to Q2), we present our revision of the proof of Theorem 9.
> > >
> > > **-Q2: Can the authors illustrate why the rate $\frac{F^0 - F}{C \alpha \eta_g \sum_{t=1}^{T} d_t} + \Phi$ is better than $\frac{F^0 - F}{C \alpha \eta_g T}$ in MoFedSAM paper?** ($F=F^*$)
> > >
> > > -A2: Sorry for the typo about the convergence rate of MoFedSAM. The convergence rate should be $\frac{F^0 - F^*}{C \alpha \eta_g T} + \Phi$ (Theorem D.4 in MoFedSAM paper). Due to the limit of the reply, we present the revised part of the proof and illustrate why FedMRUR is better in terms of convergence rate. At the beginning of the proof of Lemma 7, we define $d_t= {\sum_{i \in S_t}\Vert \triangle_i^t \Vert }/{\Vert \sum_{i \in S_t} \triangle_i^t \Vert} \ge 1$, which is obtained by **normalized aggregation**, and introduce $\varepsilon_{\delta}$ from Lemma B.1 in MoFedSAM paper. For the 2nd and 3rd term in the R.H.S of (6), we multiply them by $d_t$. Consequently, we multiply the entire R.H.S of (7) by $d_t$. Finally, we rewrite (8) and draw a new conclusion for Lemma 7:
> > > $$\mathbb{E}_t [ F(\tilde{w}^{t+1})] \le  F(\tilde{w}^t) - K \eta_g \eta_l d_t (\frac{1}{2} - 20K^2L^2\eta_l^2)\left\lVert \nabla F(\tilde{w}^t) \right\rVert^2 + K\eta_g\eta_l\left( 6K^2\eta_l^2\alpha^4\rho^2 + 5K^2\eta_l\alpha^4\rho^2\sigma^2 + 20K^3\eta_l^3\alpha^2\sigma_g^2 + 16K^3\eta_l^4\alpha^6\rho^2 + \frac{\eta_g\eta_l\alpha^3\rho^2}{N} \sigma_l^2 \right)$$
> > >
> > > Summing this new inequality over $t$ and multiplying both sides by $\frac{1}{C \alpha \eta_g \sum_{t=1}^T {d_t}}$, we have
> > >     $$\frac{1}{\sum_{t=1}^{T} d_t} \sum_{t=1}^{T} \mathbb{E} \left\lVert \nabla F(w^t) \right\rVert^2 d_t \le \frac{F^0 - F^*}{C \alpha \eta_g \sum_{t=1}^{T} d_t} + \Phi \le \frac{F^0 - F^*}{C \alpha \eta_g T} + \Phi,$$
> > > where $\Phi$ is the same as the $\Phi$ in MoFedSAM. The first term is tighter than the one in MoFedSAM since $d_t \ge 1$. There, we get the conclusion of Theorem 9 (revised version of Theorem 8). With a proper choice of $\eta_g$, $\eta_l$ and $\rho$, the convergence rate can be rewritten as $O (\frac{1}{\sqrt{SKT}}) + O \left(\frac{\sqrt{K}}{{ST}} \right) + O \left( \frac{1}{\sqrt{K}T} \right)$, where the main dominated term has the same order with the convergence rate of MoFedSAM (Theorem 4.1 in MoFedSAM paper). Our derived bound agrees with MoFedSAM in order ($O (\frac{1}{\sqrt{SKT}})$), and our convergence analysis illustrates the effect of **Normalized Aggregation** on constants.
> > >
> > > **-Q3: The first issue is the justification in the rebuttal about hyperbolic space.**
> > >
> > > -A3: In FL, the clients train local models with their own data and the server merges these models into the shared global model. The shared global model should contain as much common information from each client as possible. The global model can be seen as the tree's root, and the local model is viewed as the leaf. In addition, from the perspective of the model structure, deep neural networks can be modeled as graphs[1,2] due to their multi-layered structures containing hierarchical information. Since its volume increases exponentially with its radius, the hyperbolic space can represent hierarchies with minimal distortion[3] and reduce the number of embedding dimensions[4]. Therefore, embedding deep neural networks makes model fusion more effective and improves performance. We also conduct experiments embedding deep neural networks in different geometric spaces using multiple seeds. The results are as follows:
> > >
> > >     |  Space    | Euclidean | Hyperbolic |
> > >     | :-------: | :-------: | :--------: |
> > >     | Test Acc. |67.46(0.32)| 68.99(0.35)|
> > >
> > > From these results, we can see that embedding model in hyperbolic space improves the test accuracy of the aggregated global model.
> > >
> > > Reference:
> > > 1. Singh et.al, "Model Fusion via Optimal Transport", NeurIPS 2020
> > > 2. Liu et.al, "Deep Neural Network Fusion via Graph Matching with Applications to Model Ensemble and Federated Learning", ICML 2022
> > > 3. M. Gromov, “Hyperbolic groups”, Essays in Group Theory, 1987
> > > 4. Peng et.al, "Hyperbolic Deep Neural Networks: A Survey", IEEE Trans. PAMI 2022
> > >
> > > Thank you very much for your valuable comments, which help us to improve our work. If you have any further questions about our submission and rebuttal, please let us know.

---

> > > > ### Comment · Reviewer_EAwW · 2023-08-15
> > > >
> > > > I thank the authors for further clarifications. Let me further illustrate my points:
> > > >
> > > > 1. The $\sum_{t=1}^{T}d_t \geq T$ is indeed an interesting observation, however this also indicates that the rate of convergence in theory is the same as MoFedSAM algorithm in the worst case. Also one point I forgot to mention is that the function $F$ is the regularized function, not the original function in the FL setting in equation (1), whereas the $F$ in MoFedSAM is the original FL problem (they didn't do any regularization), therefore the convergence theory in this paper seems flawed to me.
> > > >
> > > > 2. Again I still believe that the testing accuracy comparison between Euclidean and Hyperbolic regularizations not adequate to support the claim of the authors. The hyperbolic regularization is incorporated with a neural network which may lead to some complication in parameter tuning and algorithm design for general applications. I think I could better appreciate the beauty of the regularization if there isn't a complicated neural network embedding.
> > > >
> > > > In short, I think the direction of adding regularization to improve convergence for FL is an interesting approach, yet I think this work is entangled with too many blocks where the effect of each of the block seems unclear to me. The convergence theory doesn't show a clear improvement over the SOTA.
> > > >
> > > > To AC: Unless there are more decisive arguments from the authors or other reviewers, I decide to keep my score and evaluation. I didn't see any reply from other reviewers, and I will keep an eye.

---

> > > > > ### Author Response · Authors · 2023-08-15
> > > > > **Further clarification on the theoretical results and hyperbolic regularization.**
> > > > >
> > > > > Dear reviewer, thanks for your quick reply and valuable comments, which help us a lot to improve our submission. Thanks for recognizing our theoretical contribution on improving the constant of the optimal bound and novel hyperbolic regularization for model aggregation.
> > > > >
> > > > > **-Q1:The $\sum_{t=1}^{T} d_t \ge T$ is indeed an interesting observation, however this also indicates that the rate of convergence in theory is the same as MoFedSAM algorithm in the worst case. Also one point I forgot to mention is that the function $F$ is the regularized function, not the original function in the FL setting in equation (1), whereas the $F$ in MoFedSAM is the original FL problem (they didn't do any regularization), therefore the convergence theory in this paper seems flawed to me.**
> > > > >
> > > > > -A1: Firstly, in the worst case, the convergence rates of the current first-order optimizers are all optimal including FedAgv, FedProx, MoFedSAM, etc., and the only thing that has changed for all of them is the difference in some constant terms. Theoretically, we can see that there is some improvement in the constant terms for our proposed method, as the reviewer said.  In fact, the convergence rates of FedSAM and MoFedSAM, and FedAvg are also in the same order, because all of them are optimal, and it is not possible to have an improvement in the order unless some stronger assumptions are adopted.
> > > > >
> > > > > Secondly, the regular term is due to hyperbolic regularization, not due to normalized aggregation, and we have theoretically verified the role of normalized aggregation to some extent, even if normalized aggregation is used in MoFedSAM. This contribution is also highly nontrivial and demonstrates the benefit of the normalized aggregation technique.
> > > > >
> > > > > Thirdly, we also record the values of $\frac{\sum_{t=1}^T d_t}{T}$ over different $T$ during FL training on the CIFAR100 dataset with Dirichlet 0.3 and the results are as follows:
> > > > >
> > > > >     |  T  |  100 | 200 | 300 | 400 | 500 | 600 | 700 | 800 | 900 | 1000 | 1100 | 1200 | 1300 | 1400 | 1500 | 1600 |
> > > > >     |:--: | :--: | :-: | :--:| :--:|:--: | :-: | :--:| :--:| :--:| :--: | :--: | :--: | :--: | :--: | :--: | :--: |
> > > > >     |value| 1.110|1.195|1.313|1.406|1.470|1.514|1.545|1.568|1.586| 1.601|1.613 |1.623 |1.632 |1.639 |1.645 |1.650 |
> > > > >
> > > > > From this result, we can find that $\forall T$, $\frac{\sum_{t=1}^T d_t}{t} > 1$ and  $\frac{\sum_{t=1}^T d_t}{t}$ is increasing as $T$ increasing, which also empirically verifies the improved constant of our proposed approach.
> > > > >
> > > > > **-Q2: Again I still believe that the testing accuracy comparison between Euclidean and Hyperbolic regularizations not adequate to support the claim of the authors. The hyperbolic regularization is incorporated with a neural network which may lead to some complication in parameter tuning and algorithm design for general applications. I think I could better appreciate the beauty of the regularization if there isn't a complicated neural network embedding.**
> > > > >
> > > > > -A2: First of all, we also agree with the reviewer's opinion on the philosophy of Occam's Razor in machine learning, i.e., the simpler it is, the closer it tends to be to the truth. However, it is also valuable to design appropriate algorithms according to the structure of the problem. Although hyperbolic regularization involves some complicated mathematics, it is not a disadvantage, but an advantage in model aggregation for FL with complex deep neural networks, because the neural network itself can be considered as a very high-dimensional graph, and it is difficult for the naive model aggregation to achieve neural network embedding well; while hyperbolic regularization used in our work can solve this problem well. Our experiments and ablation study can also illustrate its benefits.
> > > > >
> > > > >
> > > > > At last, thank you for your kind effort in helping us improve our submission. If you have further comments, please let us know.

---

> > > > > > ### Author Response · Authors · 2023-08-18
> > > > > >
> > > > > > Dear Reviewer EAwW:
> > > > > >
> > > > > > We really appreciate your suggestions and active discussions on our paper which largely help us improve our work. All these valuable discussions will be incorporated into our revision.
> > > > > >
> > > > > > In the rolling discussion, we have tried our maximum effort to address your concerns, but it has been a while since our last comments. Currently, we are not sure whether our response has solved your concerns and we are on the same page with you now.  If you have further suggestions, please let us know.
> > > > > >
> > > > > > Thanks again for your valuable comments and active discussions.
> > > > > >
> > > > > > Best,
> > > > > >
> > > > > > Authors.

---

> > > > > > > ### Author Response · Authors · 2023-08-20
> > > > > > >
> > > > > > > Dear Reviewer EAwW,
> > > > > > >
> > > > > > > As the discussion phase is drawing to a close, we kindly ask you whether our further explanations have addressed your concerns. If you have any further concerns, we are happy to discuss them with you.
> > > > > > >
> > > > > > > Thanks again for your valuable comments and active discussions.
> > > > > > >
> > > > > > > Sincerely,
> > > > > > >
> > > > > > > Authors.

---

> > > > > > > > ### Comment · Reviewer_EAwW · 2023-08-20
> > > > > > > >
> > > > > > > > I thank the authors for more empirical evidences.
> > > > > > > >
> > > > > > > > Summary of my mind right now:
> > > > > > > >
> > > > > > > > 1. The normalized aggregation brings interesting results both theoretically and empirically, which is a good point;
> > > > > > > >
> > > > > > > > 2. The hyperbolic regularization still seems doubtful to me. The authors provides some comparisons over the test accuracy on Cifar-100 with the Euclidean regularization but I'm still deeply concerned with this point;
> > > > > > > >
> > > > > > > > 3. I appreciate the authors effort to address my concerns with extensive numerical evidences, and the authors indeed addressed my concern with the normalized aggregation. I see that the other reviewer increased the point after the authors provided some comparison with FedCM and other algorithms;
> > > > > > > >
> > > > > > > > 4. The original paper, as I look back, was indeed not well-organized. It's a pity that I couldn't see the updated version.
> > > > > > > >
> > > > > > > > I have a mixed sentiment toward this work, that some part of it seems interesting but some part of it is doubtful, and the whole work is a mixture of different ideas. I'd say I will give a weak accept after careful consideration.
> > > > > > > >
> > > > > > > > To AC: I'll increase my point but please keep in mind that I still have some concerns regarding both the theory and the general layout of the work when you make the decision.

---

> > > > > > > > > ### Author Response · Authors · 2023-08-21
> > > > > > > > >
> > > > > > > > > Thanks for your valuable comments and suggestions, which largely help us improve our submission. In the next version, we will make the arguments more clear.  We really appreciate your effort.

---

### Official Review · Reviewer_jAXP · 2023-07-05

**Soundness:** 3 good
**Presentation:** 2 fair
**Contribution:** 2 fair
**Rating:** 5
**Confidence:** 3

**Summary:**

The authors propose a novel Federated Learning Algorithm, FedMRUR that uses hyperbolic graph diffusion to reduce the effect of data heterogeneity and thereby model inconsistencies. The authors also propose a normalized aggregation scheme to achieve faster convergence. The algorithm FedMRUR achieves state-of-the-art performance on standard datasets.



**Strengths:**

In terms of comparisons to baselines, FedMRUR seems to outperform in terms of accuracy for various Dir(\mu) settings versus the baselines. The closest competitor seems to be MoFedSAM.

**Weaknesses:**

Finally, while convergence speed is quicker for FedMRUR, Table 3 in the Supplemental Materials actually has FedCM to have the quickest total time to achieve 60% test accuracy on CIFAR-100. Do the authors have comments or suggestions on when to use FedMRUR vs FedCM given the disparity between convergence speed and clock times?


**Questions:**

1. The authors should clarify notation in Figure 1 and Algorithm 1. In Figure 1, the global parameter is subscripted, w_0, but superscripted in the algorithm.
2. The algorithm can be hard to follow as it:
  a. Did not specify the global parameter \eta_g in the parameters
  b. Did not specify the gradients, g_{i}^{t,k}, though that is easily induced
  c. More importantly did not specify how \tilde{g}_{i}^{t,k} is derived. This is mentioned later in line 213, but would make the presentation clearer by linking it directly to the equations
  d. They can also specify the normalized aggregation more clearly, and tie to the algorithm 1.
3. The authors should discuss normalized aggregation in more depth, as it seems to contribute more than hyperbolic graph fusion (Table 3). In fact, it would be interesting to understand the effect of this operator in other baselines.
4. In terms of comparisons to baselines, FedMRUR seems to outperform in terms of accuracy for various Dir(\mu) settings versus the baselines. The closest competitor seems to be MoFedSAM. First, it would be useful to have test accuracy and test accuracy std to enable better comparison between comparably performing models. Second, how was Table 1 derived, as the accuracy numbers are quite different and higher than the comparable Table 5 in the MoFedSAM paper for CIFAR-100 with Dir().
5. Similarly, the convergence table 2 seem different for MoFedSAM and FedAvg vs Table 5 in the MoFedSAM paper. Is this a setup difference?


**Limitations:**

The authors have addressed some limitations of their method

---

> ### Author Rebuttal · Authors · 2023-08-09
>
> Thank you for the constructive comments which helps us improve the paper. We have prepared our responses to each of your questions.
>
> **- Weaknesses: Finally, while convergence speed is quicker for FedMRUR, Table 3 in the Supplemental Materials actually has FedCM to have the quickest total time to achieve 60% test accuracy on CIFAR-100. Do the authors have comments or suggestions on when to use FedMRUR vs FedCM given the disparity between convergence speed and clock times?**
> - A(Weaknesses): The FedCM achieves the fastest total time is due to the less time consumption compared with FedMRUR. Because of the operations of SAM and Hyperbolic Graph Fusion scheme need a lot of computation, FedMRUR takes more time to compute in one round. For FL, academia and industry are more interested in test accuracy vs communication rounds. Under the setting of FL, the connections between devices are often through Wi-Fi or cellular network, their bandwidth is limited, and local devices are often redundant. It is common to reduce the number of communication rounds by adding local updates. In this paper to be fair, we compare test accuracy vs communication rounds with the same setting of local epochs. From the perspective of communication rounds, FedMRUR achieves the fastest convergence and the highest test accuracy; From the perspective of wall clock time, FedMRUR lags behind FedCM at low accuracy, but at higher accuracy, FedCM will gradually lag behind FedMRUR.
>
> **- Q1: The authors should clarify notation in Figure 1 and Algorithm 1. In Figure 1, the global parameter is subscripted, w_0, but superscripted in the algorithm.**
> - A1: Thank you for the careful reading. To avoid confusion, we replace '$w_0$' with $w_g$ to denote the global parameter. Please refer to Figure 1 in the attached pdf.
>
> **- Q2: The algorithm can be hard to follow as it: a. Did not specify the global parameter \eta_g in the parameters b. Did not specify the gradients, g_{i}^{t,k}, though that is easily induced c. More importantly did not specify how \tilde{g}_{i}^{t,k} is derived. This is mentioned later in line 213, but would make the presentation clearer by linking it directly to the equations d. They can also specify the normalized aggregation more clearly, and tie to the algorithm 1.**
> - A2: For the answer, please refer to Q1 in the Author Rebuttal.
>
> **- Q3: The authors should discuss normalized aggregation in more depth, as it seems to contribute more than hyperbolic graph fusion (Table 3). In fact, it would be interesting to understand the effect of this operator in other baselines.**
> - A3: For the answer, please refer to Q2 in the Author Rebuttal.
>
> **- Q4: In terms of comparisons to baselines, FedMRUR seems to outperform in terms of accuracy for various Dir(\mu) settings versus the baselines. The closest competitor seems to be MoFedSAM. First, it would be useful to have test accuracy and test accuracy std to enable better comparison between comparably performing models. Second, how was Table 1 derived, as the accuracy numbers are quite different and higher than the comparable Table 5 in the MoFedSAM paper for CIFAR-100 with Dir().**
> - A4: Thank you for the advice and the careful Experiments reading.
> First, we add some experiments to compare our method FedMRUR and MoFedSAM. The results on CIFAR100 dataset are as follow.
>
>     | Algorithm | $\mu=0.6$ | $\mu=0.3$ | $n=20$ | $n=10$ |
>     | :-------: | :-------: | :-------: | :----: | :----: |
>     | MoFedSAM  |67.51(0.37)|66.81(0.36)|64.23(0.34)| 61.25(0.40)|
>     | FedMRUR   |69.23(0.39)|68.99(0.35)|66.72(0.31)| 63.50(0.42)|
>
> In this table, we record the averaged test accuracy and the std at the 1600-th communication round over 8 seeds. From these results, we can clearly see that FedMRUR can achieves better performance than MoFedSAM.
> Second, since the authors of MoFedSAM paper don't have the open source code, we obtain the baseline based on our own implementation of the paper, and we reproduce their code as well as possible for a fair comparison. As for the experimental setup, our paper sets the customer participation rate to 0.1 and the number of local epochs to 5, while the MoFedSAM paper sets the customer participation rate to 0.2 and the number of local epochs to 10. In addition, when replicating the MoFedSAM algorithm, we employ the ASAM[1] optimizer to compute the gradient of the local loss function.
>
> **- Q5: Similarly, the convergence table 2 seem different for MoFedSAM and FedAvg vs Table 5 in the MoFedSAM paper. Is this a setup difference?**
> - A5: Thank you for the carefully reading the experiments. The difference between Table 2 in our paper and Table 5 in the MoFedSAM paper comes from the different setup in these two papers. In the MoFedSAM paper, the client participation ratio is $0.2$ and the local interval $K$ is $10$; In our paper, the client participation ration is $0.1$ and the local interval $K$ is $5$. When the data heterogeneity is not severe (Dirichlet 0.6), more participating clients $\lVert S \rVert$ and more local intervals $K$ results in a faster convergence; when the data heterogeneity is severe (Dirichlet 0.3) and larger local interval $K$ forces the local parameter to close to the local optimum, which cause severe client drifts and degrade the performance.
>
> Reference:
> 1. Kwon, et.al "ASAM: Adaptive Sharpness-Aware Minimization for Scale-Invariant Learning of Deep Neural Networks", International Conference on Machine Learning, PMLR, 2021.

---

> > ### Comment · Reviewer_jAXP · 2023-08-16
> >
> > I thank the authors for their response, which clarified questions 2-5.
> >
> > However, I am not convinced by their argument for run times in their response to Q1. I understand that communication rounds present a bottleneck, and is a standard for many FL studies, while other operations can be performed locally and asynchronously. However, does the high computation cost of SAM and Hyperbolic Graph Fusion, as stated by the authors, limit the application of FedMRUR to a more narrow range of local devices thereby limiting FedMRUR?

---

> > > ### Author Response · Authors · 2023-08-16
> > > **FedMRUR is more efficient with respect to both the communication round and wall-clock time when high-performance models are required.**
> > >
> > > Dear reviewer, thanks for your quick reply and valuable comments, which help us a lot to improve our submission.
> > >
> > > **-Q: However, does the high computation cost of SAM and Hyperbolic Graph Fusion, as stated by the authors, limit the application of FedMRUR to a more narrow range of local devices thereby limiting FedMRUR?**
> > >
> > > -A: To answer this question, here we report the communication rounds, sampled gradient, and wall-clock time used to achieve different test accuracy on CIFAR100 with FedMRUR, FedCM, and MoFedSAM.
> > >
> > >     | Rounds vs. Acc. | 40% | 45% | 50% | 55% | 58% | 60% | 61% | 62% | 63% |
> > >     | :--------------:|:---:|:---:|:---:|:---:|:---:|:---:|:---:|:---:|:---:|
> > >     |     FedCM       | 160 | 200 | 249 | 307 | 390 | 464 | 543 | 684 | 994 |
> > >     |    MoFedCM      | 177 | 217 | 258 | 315 | 364 | 403 | 447 | 492 | 573 |
> > >     |    FedMRUR      | 198 | 219 | 253 | 288 | 318 | 352 | 372 | 411 | 438 |
> > >
> > >     | Sampled Gradient (x1000) vs. Acc. | 40% | 45% | 50% | 55% | 58% | 60% | 61% | 62% | 63% |
> > >     | :-------------------------:|:---:|:---:|:---:|:---:|:---:|:---:|:---:|:---:|:---:|
> > >     |           FedCM            |  84 | 100 |124.5|153.5| 195 | 232 |271.5| 342 | 497 |
> > >     |          MoFedCM           | 177 | 217 | 258 | 315 | 364 | 403 | 447 | 492 | 573 |
> > >     |          FedMRUR           | 198 | 219 | 253 | 288 | 318 | 352 | 372 | 411 | 438 |
> > >
> > >     | Wall-clock time(s) vs. Acc.|   40%   |   45%   |   50%   |   55%   |   58%   |   60%   |   61%   |   62%   |    63%   |
> > >     | :-------------------------:|:-------:|:-------:|:-------:|:-------:|:-------:|:-------:|:-------:|:-------:|:--------:|
> > >     |           FedCM            | 2205.84 | 2626.00 | 3269.37 | 4030.91 | 5120.70 | 6092.32 | 7129.59 | 8980.92 | 13051.22 |
> > >     |          MoFedCM           | 3571.86 | 4379.06 | 5206.44 | 6356.70 | 7345.52 | 8132.54 | 9020.46 | 9928.56 | 11563.14 |
> > >     |          FedMRUR           | 4369.86 | 4833.33 | 5583.71 | 6356.16 | 7018.26 | 7768.64 | 8210.04 | 9070.77 |  9666.66 |
> > >
> > > Due to the double calculation of the gradients via SAM, MoFedSAM, and FedMRUR take more time in a single round. However, the communication rounds required are much less than FedCM. From these tables, we can find that FedMRUR requires the least wall-clock time and sampled gradients to achieve high test accuracy due to the fastest convergence of FedMRUR under the same hardware conditions. Considering the total wall-clock time costs, the acceleration ratio of FedMRUR achieves $1.20 \times$ compared with MoFedSAM ($1.35 \times$ compared with FedCM) at the final. Therefore, we can conclude that the FedMRUR is more efficient with respect to the communication round, gradient calculation, and wall-clock time when high-performance models are required.
> > >
> > >
> > > At last, thank you very much for your valuable comments, which help us to improve our work. If you have any further questions about our submission and rebuttal, please let us know.

---

> > > > ### Comment · Reviewer_jAXP · 2023-08-16
> > > >
> > > > This table is interesting as it suggests that FedMRUR eventually reaches higher test accuracy (63%) quicker than FedCM and MoFedCM, but that is only 1 data point, and it might be difficult to have more test accuracy points higher to validate the point further.
> > > > I agree the acceleration ratio appears lower.

---

> > > > > ### Author Response · Authors · 2023-08-16
> > > > > **Quick reply.**
> > > > >
> > > > > Dear reviewer, thanks for your reply.
> > > > >
> > > > > Below, we first give a quick reply to your comments.  Once we finish the full table with more target accuracy data points, we will post it as soon as possible.
> > > > >
> > > > > **-Q: This table is interesting as it suggests that FedMRUR eventually reaches higher test accuracy (63%) quicker than FedCM and MoFedCM, but that is only 1 data point, and it might be difficult to have more test accuracy points higher to validate the point further. I agree the acceleration ratio appears lower.**
> > > > >
> > > > > -A: Since during the rolling discussion stage, we can not upload the figure version of the above three tables.  In fact, we can further increase the target accuracy data point to greater than 68%.  In this setting, both the FedCM and MoFedSAM are hard to achieve this target accuracy (with more than 1500 communication rounds). The acceleration ratio will be more clear (larger than $2\times$ or even more). Once we finish the full table, we will print it to show the large acceleration ratio.
> > > > >
> > > > > From the theoretical perspective, all the FedCM, MoFedSAM, and FedMRUR are first-order optimizers with a sublinear convergence rate. Thus, they struggle to increase the accuracy at the late stage. Please check Table 1 and Figure 3 in the submission. Both FedCM and MoFedSAM are almost stagnated to achieve 65% accuracy.  While thanks to the introduced normalization and hyperbolic regularization techniques, FedMRUR can achieve a higher accuracy, which results in a high acceleration ratio in the late stage.
> > > > >
> > > > > **In summary, when higher accuracy is required,  the acceleration ratio of FedMRUR over FedCM and MoFedSAM will be much larger.**
> > > > >
> > > > > Hope this quick reply can partially solve your concern. We will also incorporate the full table in our revision.

---

> > > > > > ### Author Response · Authors · 2023-08-17
> > > > > > **Larger acceleration ratio can be achieved by FedMRUR**
> > > > > >
> > > > > > Dear reviewer, thanks for your reply. Here we present the full table with more test accuracy points.
> > > > > >
> > > > > > The table recording gradient calculations, communication rounds, and wall-clock time to achieve various accuracy ($\ge$ 62.5%) on CIFAR100 with FedMRUR, FedCM, and MoFedSAM are as follows:
> > > > > >
> > > > > >     | Rounds \ Acc.  |62.5%|62.7%|62.9%|63.1%|63.3%|63.5%|63.7%|63.9%|64.1%|64.3%|64.5%|64.7%|64.9%|65.1%|65.3%|65.5%|65.9%|66.3%|66.7%|67.1%|
> > > > > >     | :-------------:|:---:|:---:|:---:| :-: |:---:|:---:|:---:|:---:|----:|:---:|:---:|:---:|:---:| :-: | :--:|:---:| :--:|:---:|:---:|:---:|
> > > > > >     |     FedCM      | 749 | 873 | 933 | 997 | 999 | 1166| 1395| 1395| 1396| 1561| 1727| --- | --- | --- | ----| ----| ----| ----| ----| ----|
> > > > > >     |    MoFedCM     | 525 | 548 | 573 | 585 | 586 |  604|  606|  645|  662|  747|  855| 1080| 1194| 1295| 1421| 1512| 1561| 1623| 1662| ----|
> > > > > >     |    FedMRUR     | 421 | 428 | 438 | 441 | 450 |  466|  500|  502|  502|  544|  544|  559|  591|  594|  640|  652|  674|  704|  728|  786|
> > > > > >
> > > > > >     | Gradients(x1000)\Acc.|62.5%|62.7%|62.9%|63.1%|63.3%|63.5%|63.7%|63.9%|64.1%|64.3%|64.5%|64.7%|64.9%|65.1%|65.3%|65.5%|65.9%|66.3%|66.7%|67.1%|
> > > > > >     | :-------------------:|:---:|:---:|:---:| :-: |:---:|:---:|:---:|:---:|----:|:---:|:---:|:---:|:---:| :-: | :--:|:---:|:---:|:---:|:---:|:---:|
> > > > > >     |           FedCM      |374.5|436.5|466.5|498.5|499.5|  583|697.5|697.5|  698|780.5|863.5| --- | --- | --- | ----| ----| ----| ----| ----| ----|
> > > > > >     |          MoFedCM     | 525 | 548 | 573 | 585 | 586 |  604|  606|  645|  662|  747|  855| 1080| 1194| 1295| 1421| 1512| 1561| 1623| 1662| ----|
> > > > > >     |          FedMRUR     | 421 | 428 | 438 | 441 | 450 |  466|  500|  502|  502|  544|  544|  559|  591|  594|  640|  652|  674|  704|  719|  756|
> > > > > >
> > > > > >     |time(s) vs. Acc.|  62.5% |  62.7% |  62.9% |  63.1% |  63.3% |  63.5% |  63.7% |  63.9% |  64.1% |  64.3% |  64.5% | 64.7%  | 64.9%  | 65.1%  |  65.3% |  65.5% |  65.9% |  66.3% |  66.7% |67.1%|
> > > > > >     |:--------------:|:------:|:------:|:------:|:------:|:------:|:------:|:------:|:------:|:------:|:------:|:------:|:------:| :----: | :----: | :-----:|:------:|:------:|:------:|:------:|:---:|
> > > > > >     |     FedCM      | 9821.40|11462.49|12250.29|13090.61|13116.87|15309.58|18316.35|18316.35|18329.48|20487.56|22664.28|  ----  | -----  | ------ | -------| -------| -------| -------| -------| ----|
> > > > > >     |    MoFedCM     |10594.50|11058.64|11563.14|11805.30|11825.48|12188.72|12229.08|13016.10|13359.16|15063.29|17235.73|21782.48|24083.88|26132.94|28657.26|30512.16|31500.98|32752.14|33539.16|--|
> > > > > >     |  FedMRUR       | 9291.47| 9445.96| 9666.66| 9717.53| 9921.31|10284.62|11035.00|11079.14|11079.14|12001.36|12001.36|12327.16|13024.54|13044.84|14124.25|14389.64|14949.32|15614.72|15947.04|16768.08|
> > > > > >
> > > > > > From this table, we can find that FedMRUR takes the least wall-clock time, and the acceleration ratio increases from $1.05 \times$ (62.5% accuracy) to $1.88 \times$ (64.5% accuracy) compared with FedCM and FedCM can't achieve 64.7% accuracy. Compared with MoFedSAM, the acceleration ratio increases from $1.05 \times$ (60% accuracy) to $2.10 \times$ (66.7% accuracy) and MoFedSAM can't achieve 67.1% accuracy. We can conclude that because of the normalized aggregation and hyperbolic regularization techniques, FedMRUR can accelerate the training time and achieve high test accuracy.
> > > > > >
> > > > > > **These tables further verify that when higher accuracy is required, the acceleration ratio of FedMRUR over FedCM and MoFedSAM will be much larger. We hope this verification can solve your concerns completely**
> > > > > >
> > > > > > At last, thank you very much for your valuable comments, which help us to improve our work. If you have any further questions about our submission and rebuttal, please let us know.

---

> > > > > > > ### Comment · Reviewer_jAXP · 2023-08-17
> > > > > > >
> > > > > > > I think these empirical results are interesting and am willing to raise to grade accordingly

---

> > > > > > > > ### Author Response · Authors · 2023-08-18
> > > > > > > >
> > > > > > > > Dear Reviewer jAXP,
> > > > > > > >
> > > > > > > > We wish to express our sincere gratitude for your valuable insights and comments which largely help us improve our work.
> > > > > > > > Your support is greatly appreciated!
> > > > > > > >
> > > > > > > > Best regards,
> > > > > > > >
> > > > > > > > Authors

---

### Official Review · Reviewer_aqu4 · 2023-07-06

**Soundness:** 3 good
**Presentation:** 3 good
**Contribution:** 3 good
**Rating:** 7
**Confidence:** 3

**Summary:**

This paper studies the problem of model inconsistency across clients in federated learning (FL). The authors propose a method called FedMRUR, which uses a hyperbolic graph manifold regularization term and a normalized update aggregation scheme to alleviate the issues introduced by model inconsistency. Compared with the previous works, the proposed FedMRUR can reflect the model bias better in a low-dimensional subspace and mitigate the norm reduction of global updates caused by model inconsistency. The authors prove the convergence of FedMRUR for nonconvex objectives under partial client participation. They also run experiments to show that FedMRUR can achieve a new state-of-the-art (SOTA) accuracy with less communication.

**Strengths:**

1.	The work is written clearly and intuitively.

2.	The considered problem is quite meaningful as the model inconsistency severely impairs the performance of FL and building an algorithm to solve this issue is a promising direction to improve FL.

3.	The theoretical result is rigorous for non-convex settings under partial client participation.

4.	There are experiments to support the theory as well as to show that the algorithm will be useful in practice.


**Weaknesses:**

See questions

**Questions:**

1.	It is not clear what is the “manifold structure of their representations”.

2.	What does it mean by “norm reduction” and why it is important?

3.	For the non-i.i.d pathological-n setting on TinyImageNet, line 264 conflicts with Figure 3(b).

4.	Does the setting of the parameter $\beta$ for hyperbolic graph manifold regularization have an impact on the performance of the algorithm?

5.	In the ablation study, the author should discuss the hyperparameters sensitivity of $\gamma$ in a wider range.


**Limitations:**

Please refer to the Questions.

---

> ### Author Rebuttal · Authors · 2023-08-09
>
> Thank you for the constructive comments which helps us improve the paper. We have prepared our responses to each of your questions.
> - Q1: It is not clear what is the “manifold structure of their representations”.
> - A1: In our paper, we design the hyperbolic graph fusion scheme to mitigate the model inconsistency among the clients. In this scheme, we map the data representations of the clients to the vectors of Lorentz model in hyperbolic space. The induced distance of the Lorentz model is used to measure model inconsistency in our method and taken as the regularization term in the objective function to constrain the model inconsistency for improving the performance of FL. Since that hyperbolic geometry is a Riemannian manifold with a constant negative curvature, the typical geometric property of hyperbolic space is that its volume increases exponentially in proportion to its radius, whereas the Euclidean space grows polynomially. The hyperbolic space has 2 benefits enabling it to well deal with the complex real-world FL applications.
>     1. Hyperbolic space exhibits minimal distortion and fits the hierarchies particularly well since the space closely matches the growth rate of tree-like data while the Euclidean space cannot.
>     2. Even though with a low-embedding dimension space, hyperbolic models are surprisingly able to produce high-quality representation, which makes it to be especially favorable in low-memory and low-storage scenarios.
>
>     In realistic scenarios, there extensively exists many tree-like data structure, such as the hypernym structure in natural languages, the subordinate structure of entities in the knowledge graph, the organizational structure for financial fraud, and the power-law distribution in recommender systems. In FL, clients and the server logically also have a tree-like hierarchical topological relationship.
>     Therefore, adopting the Lorentz metric of hyperbolic space makes use of the hierarchical information in the datasets and the hierarchical relationship between the server an clients, which are helpful to group data samples further bring prediction gains.
>     We also conducts experiments using different geometric spaces over several seeds and the results are as follow.
>
>     |  Space    | Euclidean | Hyperbolic |
>     | :-------: | :-------: | :--------: |
>     | Test Acc. |    67.46(0.32)     |    68.99 (0.34)      |
>
>     From this results, we can find that using the Lortenz metric of hyperbolic space can help the algorithm achieving the highest test accuracy. With the use of "manifold structure of their representations" in hyperbolic space, FedMRUR improves the performance of FL.
> - Q2: What does it mean by “norm reduction” and why it is important?
> - A2: In FL framework, because of the data heterogeneous among the clients, there exits great divergences among the gradients of the local loss function and the similarity between the local updates are low during FL training. From the perspective of geometry, the low similarity means the vectors are almost orthogonal (i.e. the $\cos \theta \approx 0$, where $\theta$ is the angle between the vectors). When adopting the naive aggregation method, the server takes an average of many  vectors as the global update, whose norms will be very small. Then the global update becomes small, which leads to slow convergence and diminishing returns. We also conduct experiments in Ablation Study to vertify that compensating the "norm reduction" can imporve the performance of the algorithm. Therefore, it is critical to compensate the "norm reduction" in FL.
> - Q3: For the non-i.i.d pathological-n setting on TinyImageNet, line 264 conflicts with Figure 3(b).
> - A3: Thank you for the careful reading. For the non-i.i.d pathological-n setting on TinyImageNet, the coefficient $n$ should be selected from $\{40 ,80\}$, we correct this typo in the paper.
> - Q4: Does the setting of the parameter $\beta$ for hyperbolic graph manifold regularization have an impact on the performance of the algorithm?
> - A4: Thank you for the suggestion to study the effects of parameter $\beta$ on the performance. We conduct the experiemnt with various settings of $\beta$.
> Here, we provide the test accuracy after the $1600-$ th communication round.
>     | $\beta=0.1$ | $\beta=0.5$ | $\beta=1.0$ | $\beta=5.0$ | $\beta=10.0$ |
>     | :---------: | :---------: | :------:    | :------:    | :-------:    |
>     |    68.67    |    69.03    |   68.96     |   68.26     |     69.10    |
>
>    From this table, we can find that $\beta$ has a limited impact on the performance of the algorithm.
> - Q5: In the ablation study, the author should discuss the hyperparameters sensitivity of $\gamma$ in a wider range.
> - A5: Thank you for the suggestion about investigating the impact of $\gamma$ on the perofrmance. We conduct the experiemnt with various settings of $\gamma$.
> Here, we provide the test accuracy after the $1600-$ th communication round.
>     | $\gamma=0.001$ | $\gamma=0.0005$ | $\gamma=0.0002$ | $\gamma=0.0001$ | $\gamma=0.00005$ |
>     | :------------: | :-------------: | :-------------: | :-------------: | :--------------: |
>     |     63.35      |      65.95      |       67.75     |       68.54     |      68.96       |
>
>     From this table,  we find that $\gamma$ has great impacts on the performance. From (3) and (5), a high $\gamma$ slows down the local training process; a low gamma value can't support the regularization term to constrain model inconsistency. When $\gamma=0.00005$, the algorithm achieves the highest test accuracy.

---

> > ### Author Response · Authors · 2023-08-18
> >
> > Dear Reviewer aqu4:
> >
> > We really appreciate your constructive opinions that helped us improve this paper. All the discussions will be incorporated into our revision. If there are any concerns unresolved,  please let us know and we are ready to have further discussions with you.
> >
> > Thanks again for your time.
> >
> > Best,
> >
> > Authors.

---

### Official Review · Reviewer_HiX9 · 2023-07-06

**Soundness:** 3 good
**Presentation:** 3 good
**Contribution:** 3 good
**Rating:** 7
**Confidence:** 3

**Summary:**

The authors found the existing vanilla distillation in FL, the model inconsistency caused by the local data heterogeneity across clients results in the near-orthogonality of client updates, which leads to the global update norm reduction and slows down the convergence. Moreover, the authors argue previous works may fail to reflect the model inconsistency due to the complex structure of the machine learning model and the Euclidean space’s limitation in meaningful geometric representations. To resolve the above issues, they propose the FedMRUR algorithm for FL. By adopting a hyperbolic graph manifold regularizer and aggregating the client updates norms as the global update norm, the FedMRUR achieves a new state-of-the-art (SOTA) accuracy with less communication.

**Strengths:**

originality: Exploiting the manifold structures of the representations can reflect the model bias better than the parameters (or gradients) method, to significantly reduce the model inconsistency. Aggregating the client updates norms as the global update norm can mitigate the norm reduction caused by model inconsistency.

quality: The paper is well written, with detailed experiments and ablation studies with other methods and the proposed variants.

clarity: The paper consists of an illustration of the workflow and text explanations of the proposed FedMRUR algorithm.

significance: The paper solves the problem of the model inconsistency across the clients in FL, reduces the model bias more effectively than its baseline, mitigates the norm reduction caused by model inconsistency, and improves the test performance.


**Weaknesses:**

- There are some unclear points and confusing notations.
- More experiments can be conducted.


**Questions:**

1) Some notations are confusing. For example, in line 14 of Algorithm 1, what does the notation “$\delta_i^{t}$” mean?
2) In line 263, what should the setting of the Dirichlet coefficient be?
3) The authors should discuss the effect of local interval $K$ on the performance of the algorithm.


**Limitations:**

Please find the weaknesses and limitations.

---

> ### Author Rebuttal · Authors · 2023-08-09
>
> Thank you for the constructive comments which helps us improve the paper. We have prepared our responses to each of your questions.
> - Weakness
>     **- W1: There are some unclear points and confusing notations.**
>     - A1: Thank you for the careful readings. We check the whole paper and correct all the typos. We eliminated the unclear points and confusing notations. We unify the notations in Figure 1, Algorithm 1 and Section 3.2. In Figure 1, we replace $w_0$, $F_0$ with $w_g$, $F_g$ to denote the global parameter and function. Please refer to Figure 1 in the attached pdf. In Algorithm 1, the local gradient is denoted by $\nabla F_p(\cdot)$, where $F_p(\cdot)$ is the local loss function formulated by (5) in Section 3.2. We replace Figure 2 with a new figure (Figure 2 in the attached pdf) to present the "Normalized Aggregation of Local Updates" more clearly. We also fix some typos in the proof. The convergence rate is $O \left(\frac{1}{\sqrt{SKT}}\right) + O \left(\frac{\sqrt{K}}{{ST}}\right) + O \left(\frac{1}{\sqrt{K}T}\right)$ in Theorem 1. In this way, we clearly demonstrate the two components of FedMRUR: "hyperbolic graph fusion" and "normalized aggregation of local updates" and validate the effectiveness of the algorithm.
>
>     **- W2: More experiments can be conducted.**
>     - A2: Thank you for the advice about adding experiments.
>     We conduct the experiment on discussing the effect of local interval $K$ and validate the linear speedup property of FedMRUR. The results can be seen from the Figure 3 in the attached pdf. From the figure, when $K$ increases to $8$, the algorithm performs $2.0\times$ faster than $K=4$. When $K$ is increased to $\mathcal{O} (ST)^{\frac{1}{2}}$, the impact of the second dominated term of (7) in Theorem 1 becomes greater and the first term becomes less. When $K$ increase from 8 to 16, the speedup of convergence is not obvious.
>     We also extend the range of $\gamma$ to study its hyper-parameter sensitivity by experiments.
>
>         | $\gamma=0.001$ | $\gamma=0.0005$ | $\gamma=0.0002$ | $\gamma=0.0001$ | $\gamma=0.00005$ | $\gamma=0.00002$ |
>         | :------------: | :-------------: | :-------------: | :-------------: | :--------------: | :--------------: |
>         |     63.35      |      65.95      |       67.75     |       68.54     |      68.96       |      68.45       |
>
>         Form this table,  we find that $\gamma$ has great impacts on the performance. From (3) and (5), a high $\gamma$ slows down the local training process; a low gamma value can't support the regularization term to constrain model inconsistency. When $\gamma=0.00005$, the algorithm achieves the highest test accuracy.
>         To study the impact of $\beta$ for hyperbolic graph manifold regularization on the performance, we conduct the experiment on CIFAR100 task with different $\beta$ settings and present the final test accuracy as follows.
>
>         | $\beta=0.1$ | $\beta=0.5$ | $\beta=1.0$ | $\beta=5.0$ | $\beta=10.0$ |
>         | :---------: | :---------: | :------:    | :------:    | :-------:    |
>         |    68.67    |    69.03    |   68.96     |   68.26     |     69.10    |
>
>         From this table, we can find that $\beta$ has a limited impact on the final performance of the algorithm.
> - Questions
>     **- Q1: Some notations are confusing. For example, in line 14 of Algorithm 1, what does the notation “$\delta_i^t$” mean?**
>     - A1: Sorry about the unclear presentation. "$\delta_i^t$" means the accumulated update of the local parameter at client $i$ within $t-$ th round (i.e., $w_i^{t,K} - w_i^{t,0}$). We have fix this issue in Algorithm 1.
>
>     **- Q2: In line 263, what should the setting of the Dirichlet coefficient be ?**
>     - A2: Thank you for the careful Experiments readings. The setting of the Dirichlet coefficient should be $\{0.3, 0.6\}$.
>
>     **- Q3: The authors should discuss the effect of local interval $K$ on the performance of the algorithm.**
>     - A3: Thank you for the advice about experiments. We conduct the experiment with different local interval $K$ on CIFAR-100 dataset and plot the test accuracy in communication rounds. In a certain range, enlarging $K$ can accelerate the convergence and improve the performance. When $K$ is out of the range, larger $K$ means more updates on the local dataset, which forces the local parameter $w_{p}^{t,k}$ closed to local optimum $x_{p}^*$. This causes severe client drifts that degrade the performance. In addition, since the number of local datasets changes as the number of clients grows, we also test the linear speedup property by changing $K$. From the figure, when $K$ increases to $8$, the algorithm performs $2.0\times$ faster than $K=4$. When $K$ is increased to $\mathcal{O} (ST)^{\frac{1}{2}}$, the impact of the second dominated term of (7) in Theorem 1 becomes greater and the first term becomes less. When $K$ increase from 8 to 16, the speedup of convergence is not obvious.

---

> > ### Author Response · Authors · 2023-08-18
> >
> > Dear Reviewer HiX9:
> >
> > We really appreciate your constructive opinions that helped us improve this paper. All the discussions will be incorporated into our revision. If there are any concerns unresolved, we would be glad to have further discussions.
> >
> > Thanks again for your time.
> >
> > Best,
> >
> > Authors.

---

### Official Review · Reviewer_mqsa · 2023-07-26

**Soundness:** 3 good
**Presentation:** 2 fair
**Contribution:** 3 good
**Rating:** 5
**Confidence:** 3

**Summary:**

This paper proposes a federated learning framework called FedMRUR to deal with the model inconsistency caused by the local data heterogeneity across clients and insufficient geometric representation ability. To do this, it adopts a hyperbolic graph manifold regularizer to ensure that the representations obtained by the local and global models are close in a low-dimensional subspace. Then it aggregates the client updates norms as the global update norm to mitigate the norm reduction introduced by the near-orthogonality of client updates. A linear speedup property is theoretically proved for the proposed algorithm.

**Strengths:**

+ The motivation to deal with the model inconsistency and insufficient representation ability is clear.
+ It is interesting to adopt the hyperbolic graph fusion scheme in federated learning.
+ The theoretical guarantee is provided for the convergence and especially the proposed method exhibits a linear speedup.
+ The experimental results seem promising.

**Weaknesses:**

I'm not very familiar with federated learning. Here are some major concerns:
- There are many ways to exploit the manifold structure. What's the advantage of adopting the graph fusion scheme, especially in the hyperbolic space? If the hyperbolic space is a good choice, then why not let the whole learning process work in the hyperbolic space?
- Figure 2 is less informative since the readers cannot understand how the normalized aggregation works and what's the difference between these two operations in terms of their workflow.
- It seems that the widely used CIFAR-10 benchmark is not involved for evaluation.
- The experiments to validate the linear speedup property are missing.

**Questions:**

Please see the above part.

---

> ### Author Rebuttal · Authors · 2023-08-09
>
> Thank you for the constructive comments which helps us improve the paper. We have prepared our responses to each of your questions.
> - Weakness:
>     **- W1: There are many ways to exploit the manifold structure. What's the advantage of adopting the graph fusion scheme, especially in the hyperbolic space? If the hyperbolic space is a good choice, then why not let the whole learning process work in the hyperbolic space?**
>     - A1：For the answer, please refer to Q1 in the Author Rebuttal.
>
>     **- W2: Figure 2 is less informative since the readers cannot understand how the normalized aggregation works and what's the difference between these two operations in terms of their workflow.**
>     - A2: Sorry for the unclear presentation of Figure 2. Figure 2 is a toy schematic to compare the naive aggregation and normalized aggregation of the local updates. The main difference is the norm $\lVert \triangle \rVert$ of the global update, which is the aggregation of the local updates $\triangle_p$ from the participated clients. When adopting the naive aggregation method, the server directly takes the average of the local updates as the global update. Each client's contribution on the global update is $\lVert \triangle_p \rVert \cos \theta_p$, where $\theta_p$ is the angle between the local update $\triangle_p$ and the global update $\triangle$. When the data heterogeneous is significant, the divergence among the directions of local updates is large. Correspondingly, the divergence between the directions of each local update and the global update is large, and the global update norm shrinks which slows down the convergence of the global objective function. When adopting the normalized aggregation method, the server computes the direction of the global update by normalizing the sum of local updates, which is the same as the naive method. The global update norm is obtained by averaging the local updates. In this way, the client $p$'s contribution on the global update $\theta$ grows from $\lVert \triangle_p \rVert \cos \theta_p$ to $\lVert \triangle_p \rVert$. Accordingly, the norm of the global update grows and the speedup the convergence. In order to compare the two aggregation methods, we replace Figure 2 with a new one (Figure 2 in attached pdf).
>
>     **- W3: It seems that the widely used CIFAR-10 benchmark is not involved for evaluation.**
>     - A3: We place more challenged results on CIFAR100 and TinyImageNet in the amin part of the paper. The results on CIFAR-10 are presented in Figure 1, Table 1 and 2 of the Supplementary Material. From these results, FedMRUR achieves the fastest convergence and the highest test accuracy.
>
>     **- W4: The experiments to validate the linear speedup property are missing.**
>     - A4: Thank you for the reminder about validating the linear speedup property. Because the whole dataset is fixed, increasing the number of clients changes the amount of data in the local data, which changes the entire optimization problem, we conduct the experiment under various settings of local intervals $K$ fixing the number of clients to verify the linear speedup property. From Figure 3 in the attached pdf, when $K$ increases to $8$, the algorithm performs $2.0\times$ faster than $K=4$. When $K$ is increased to $\mathcal{O} (ST)^{\frac{1}{2}}$, the impact of the second term of (7) in Theorem 1 becomes greater and the first term becomes less. When $K$ increase from 8 to 16, the speedup of convergence is not obvious.

---

> > ### Author Response · Authors · 2023-08-18
> >
> > Dear Reviewer mqsa:
> >
> > We really appreciate your constructive opinions that helped us improve this paper. If there are any concerns unresolved, we would be glad to have further discussions.
> >
> > Thanks again for your time.
> >
> > Best,
> >
> > Authors.

---

### Author Rebuttal · Authors · 2023-08-10

**Thank you for the constructive comments which helps us improve the paper. We have prepared our responses to the common questions.**

**-Q1: Problem on hyperbolic space and presentation of algorithm. (EAwW;jAXP;mqsa)**
- A1:Sorry for our unclear presentation, we describe our method more clearly as follows:
    - The "mapped Lorentzian vectors" $L$ of (3) are the model representation $Z$ mapped in the hyperbolic space. In our method, we take the distance $R(w_p,w_g)$ of the mapped Lorentzian vectors in hyperbolic space defined by (4) and (5)  to measure the bias between model $w_p$ and $w_g$. This distance is added to the original loss function $f(\cdot)$ as a regularization term to constrain the model inconsistency between the client and server side. Algorithm 1 solves the new regularized $F(\cdot)$. To illustrate the connection between Algorithm 1 and Section 3.2 clearly, we fix the typos in Algorithm 1.
    - Since hyperbolic geometry is a Riemann manifold with a constant negative curvature, its typical geometric property is that the volume grows exponentially with its radius, whereas the Euclidean space grows polynomially.
    Such a geometric trait has 2 advantages:
        1. The hyperbolic space exhibits minimal distortion and it fits the hierarchies particularly well since the space closely matches the growth rate of tree-like data while the Euclidean space doesn't.
        2. Even with a low-embedding dimensional space, hyperbolic models are surprisingly able to produce a high quality representation, which makes them to be particularly advantageous in low-memory and low-storage scenarios.

        In realistic scenarios, there exists many tree-like data structure, such as the hypernym structure in NLP, the subordinate structure of entities in the knowledge graph and the power-law distribution in recommender systems. In FL, clients and the server logically also have a tree-like hierarchical topological relationship, so adopting the Lorentz metric of hyperbolic space makes use of the hierarchical information in the datasets and the hierarchical relationship between the server and clients, which are helpful to group data samples further bring prediction gains. Using Euclidean metric, or some Riemann metric defined by a positive definite matrix is an interesting idea. Here, we show the results of experiments using different geometric spaces as follow.

        |  Space    | Euclidean | Hyperbolic |
        | :-------: | :-------: | :--------: |
        | Test Acc. |    67.46(0.32)     |    68.99(0.35)        |

        From these results, we can find that Lortenz metric of hyperbolic space can help the algorithm achieving the highest test accuracy.  The reason is that the traits of hyperbolic space make good use of the hierarchical information in the datasets, which is helpful to group data samples and further bring prediction gains.

        The representations generated by the model have fewer dimensions than the data. Mapping the representations to the hyperbolic space introduces less computation overhead than mapping the data. We also conduct experiments mapping the original data to the hyperbolic space over 8 seeds. The results are as follow.

        | Original data | Representations |
        |     :---------:   |   :---------:   |
        | 69.05(0.57) |   68.99(0.35)   |

        From the table, we can see that both methods achieve similar performance.Thus, considering the computation overhead and performance, we only map representations to hyperbolic space and do not treat the entire learning process in hyperbolic space.
    - Section 3.2 is the basis of Algorithm 1. The first component is **Hyperbolic Graph Fusion**. With this scheme, we obtain a new manifold regularization term $R(w_p, w_g)$ to constrain the model inconsistency and formulate the new objective function $F(\cdot)$ with the regularization term. All the gradient computations are about $F(\cdot)$, such as '$g_{i}^{t,k}$'. To depict our method more clearly, we replace '$g_{i}^{t,k}$' with '$\nabla F(\cdot)$' to denote the gradient.
    The second component is **Normalized Aggregation of Local Updates**, a new global optimizer to compensate for the global update norm reduction, as described in line 14 of Algorithm 1.

**- Q2: Problem on Normalized Aggregation (EAwW;jAXP)**
- A2: Thank you for the comment on "Normalized Aggregation of Local Updates" from the perspective of theoretical rate.
We check the mathematical part and correct some typos in the proof of Lemma 8 and Theorem 9. The main dominated term of convergence rate is ${(F^0 - F^*)}/{(C \alpha \eta_g \sum_{t=1}^{T} d_t)} + \Phi$ in Theorem 9 (new version), where $d_t = {\sum_{i \in S_t}\Vert \triangle_i^t \Vert }/{\Vert \sum_{i \in S_t} \triangle_i^t \Vert}$ is introduced by "Normalized Aggregation of Local Updates". Compared with the main dominant term in convergence rate of MoFedSAM (${(F^0 - F^*)}/{(C \alpha \eta_g T)}$, in Theorem D.7 of the paper), the convergence rate of FedMRUR is faster than MoFedSAM. Because $\forall t, d_t \ge 1$ derived from the triangle inequality. Therefore, From the theoretical view, we can conclude that the "Normalized Aggregation of Local Updates" can accelerate the convergence. In fact, using this operator in other baselines can also improve the performance. Here, we present the effect of the normalized aggregation method applied to FedCM.
  | Algorithm | $\mu=0.6$ | $\mu=0.3$ | $n=20$ | $n=10$ |
    |:---------:|:---------:|:---------:|:------:|:------:|
    | FedCM+    |  66.23(0.45)   |   65.23(0.48)   |  61.97(0.43) |  59.27(0.42)  |
    | FedCM     |   64.87(0.42)   |   63.50(0.44)   |  60.58(0.39) |  57.56(0.36)  |
    | FedAvg+   |  55.24(0.43)   |   53.05(0.41)   |  46.89(0.38) |  42.14(0.37) |
    | FedAvg    |  53.85(0.38)   |   51.20(0.37)   |  45.70(0.32) |  40.80(0.33) |

---

### Decision · Program_Chairs · 2023-09-21

**Decision:**

Accept (poster)

**Comment:**

All reviewers are unanimously positive about this work although some of them raised concerns. The main strength of this paper includes technical novelty, theoretical justification, and comprehensive evaluation. On the other hand, the reviewers generally agree that the presentation could be improved. Overall, since the strengths outweigh the weaknesses, this is good enough to be presented in NeurIPS this year.